

# Spawner weight and ocean temperature drive Allee effect dynamics in Atlantic cod, *Gadus Morhua*: inherent and emergent density regulation

Anna-Marie Winter [1*], Nadezda Vasilyeva [2], Artem Vladimirov [2,3]

[1]Centre for Ecological and Evolutionary Synthesis, Department of Biosciences, University of Oslo, 0316 Oslo, Norway
[2]Interdisciplinary Laboratory for Mathematical Modeling of Soil Systems, V.V. Dokuchaev Soil Science Institute, Pyzhevsky per. 7/2, 119017 Moscow, Russian Federation
[3]Bogoliubov Laboratory of Theoretical Physics, Joint Institute for Nuclear Research, Joliot-Curie 6, 141980 Dubna Russian Federation
*currently at Wageningen Marine Research, Wageningen University and Research, Ijmuiden, Netherlands

*Correspondence to*: Anna-Marie Winter (anna-marie.winter@wur.nl)

**Abstract.** Stocks of Atlantic cod, *Gadus Morhua*, show diverse recovery responses when fishing pressure is relieved. The expected outcome of reduced fishing pressure is that the population regains its size. However, there are also cod stocks that seem to be locked in a state of low abundance from which population growth does not, or only slowly, occur. A plausible explanation for this phenomenon can be provided by the Allee effect, which takes place when recruitment per capita is positively related to population density or abundance. However, because of methodological limitations and data constraints, such a phenomenon is often perceived as being rare or non-existent in marine fish.

In this study, we used time-series of 17 Atlantic cod stocks to fit a family of population equations that consider the abundance of spawners, their body weight as well as sea water temperature as independent components of recruitment. The developed stock-recruitment function disentangles the effects of spawner abundance, spawner weight and temperature on recruitment dynamics and captures the diversity of density dependencies (compensation, Allee effect) of the recruitment production in Atlantic cod.

The results show for 13 cod stocks an inherent, spawner abundance related Allee effect. Allee effect strength, i.e. the relative change between maximum and minimum recruitment per capita at low abundance, was *increased* when recruitment production was suppressed by unfavourable changes in water temperature and/or in spawner weight. The latter can be a concomitant of heavy fishing or a result of temperature related altered body growth. Allee effect strength was *decreased* when spawner weight and/or temperature elevated recruitment production. We show how anthropogenic stress can increase the risk of Allee effects in stocks where ocean temperature and/or spawner weight had been beneficial in the past, but are likely to "unmask" and strengthen an inherent Allee effect under future conditions.



## 1 Introduction

Almost every ecosystem is affected by rising anthropogenic pressure (Jones et al., 2018). Climate change, fishing and pollution put the ocean under an increasing level of stress. In order to secure global food production from wild fisheries, scientists and policy makers are therefore shifting their focus to the rebuilding of depleted fish stocks (Costello et al., 2020). Fast growth at low abundance gives a population the ability to recover from disturbances and make it resilient to environmental and anthropogenic alterations (Dulvy et al., 2004; Mace et al., 2008; Lande, 1994). Thus, the premise behind most fisheries management strategies and recovery plans is that the primary factor inhibiting recovery is fishing. However, several marine fish stocks show surprisingly little response to restrictions of fishing pressure (Hutchings, 2015a; Hutchings and Reynolds, 2004) and some stocks, such as the Northern cod, remain depleted since decades despite a commercial fishing moratorium (Dfo, 2019). Further, traditional assumptions about stationarity and stability of fisheries production from the oceans is challenged by changes in productivity regimes in the majority of fish stocks (Britten et al., 2017; Hilborn et al., 2014; Vert-Pre et al., 2013), which makes predictions about fisheries recovery difficult.

Capturing and understanding the abundance and dynamics of marine fish at low abundance is challenged by a scarce data availability (Pepin, 2016) and most global biomass data sets are based on stock assessment models (Ricard et al., 2012). Some stock assessment models consider recruitment -juveniles old enough to be caught in a fishery- as a stochastic process (Nielsen and Berg, 2014). In other assessment models, the production of juveniles is simulated with a stock-recruitment model which relates recruitment to spawning stock biomass (SSB)- the biomass of adult and mature females- in a function (Ricker, 1954; Beverton and Holt, 1957). The functional form for per capita production is then considered linear, because log-transformed the relationship can be linearized. SSB is the product of spawner abundance and spawner weight and is taken as a proxy for the stock's reproductive potential in general. It generally does not consider any mechanisms of non-linearity or shifts in the linkage between reproductive potential and SSB and its components (i.e. weight, probability of maturation). Also environmental variables, if included in stock-recruitment models, are usually considered as linear, density-independent terms (e.g.(Stiasny et al., 2016; Gröger et al., 2009; Stige et al., 2013; Ottersen et al., 2013)). Non-linear interactions between the different factors of the stock-recruitment model can mask or mitigate the recruitment prediction, which can affect and limit the perceived prediction of the stock's trajectory (Devine et al., 2014; Glaser et al., 2014). Thus, the urgent question remains how the different components of the stock-recruitment model, individually and together, affect recruitment production, and in particular at low abundance.

A potential mechanism that can lead to a productivity shift is the Allee effect (Jiang and Shi, 2010; Dai et al., 2012; Dakos et al., 2012). In depleted fish populations, the Allee effect has been suggested as an explanation for the observed lack of recovery when fishing was substantially reduced (Hutchings, 2001; Hutchings, 2015b). The Allee effect is an inverse density-



dependence between the per capita growth rate and population abundance or density when the population is small (Courchamp et al., 1999; Hutchings, 2015b). In contrast to the compensatory per capita growth rate caused by the release from density-dependent control, the per capita growth rate drops below the Allee effect threshold, leading to a decline in population productivity. For terrestrial populations, Allee effects have been intensively studied (Armstrong and Wittmer, 2011; Vortkamp et al., 2020; Courchamp et al., 2008), but in marine fish many (meta-) studies have failed to find a statistically significant Allee

effect, because of scarce data at low abundance or poor statistical methodology (Perälä and Kuparinen, 2017; Liermann and Hilborn, 1997; Myers et al., 1995; Hilborn et al., 2014; Sibly et al., 2005; Gregory et al., 2010). As a result, there remains a great deal of uncertainty surrounding the prevalence of Allee effects in fishes.

Alterations in population productivity can also be caused when fishing changes the population's demography and targets specific phenotypic and productivity related traits (Enberg et al., 2009; Trippel, 1995; Beamish et al., 2006). Fishing often

targets the largest, oldest individuals and when the remaining younger fish do not have the same productivity per unit biomass as older fish, for example because smaller and younger fish are less fertile and less experienced spawners, the population shifts to a lower productivity regime (Hutchings, 2005; Murawski et al., 2001; Marteinsdottir and Thorarinsson, 1998; Beamish et al., 2006). Under a short-tailed age/size structure, the population can rely less on the increased quantity and quality of reproductive output by older, experienced and large fish (Marteinsdottir and Steinarsson, 1998; Hixon et al., 2013; Birkeland

and Dayton, 2005; Murawski et al., 2001). In contrast, population productivity can be increased when strong and persisting selective fishing results in an evolutionary adaptation to high mortality, leading to an evolutionary pressure towards faster life histories, such as earlier maturation, reduced post-maturation growth and increased reproductive investment (Nussle et al., 2016; Dunlop et al., 2015). Stronger devotion to reproductive output can, however, also increase the survival cost of reproduction, resulting in higher natural mortality, a shorter life span and reduced production of recruitment per spawner life

(Kuparinen et al., 2012). For Atlantic cod, *Gadus Morhua*, there is strong evidence that targeting the largest and oldest individuals leads to an age-size truncation (e.g. (Svedäng and Hornborg, 2017; Law, 2000; Ottersen, 2008; Shelton et al., 2015)), juvenation and a productivity change (Dunlop et al., 2015; Heino et al., 2015; Sharpe and Hendry, 2009; Ottersen et al., 2014; Svedäng and Hornborg, 2017).

Temperature can also influence recruitment production, indirectly by changing body growth and food availability, and directly

by affecting egg and juvenile survival, which has led to a skewed population demography of smaller sized individuals (Cheung et al., 2013; Daufresne et al., 2009; Tu et al., 2018) with repercussions for population productivity and resilience. Temperature change has the potential of altering Allee effect dynamics, for example by suppressing recruitment production (Winter et al., 2020) or increasing adult mortality (Berec, 2019). Therefore, if strong stock depletion is accompanied by changes in spawner weight and sea temperature, recruitment dynamics are likely to change and patterns of an Allee effect could emerge, be



strengthened or masked. This questions the common assumption that Allee effects are time-invariant, but requires a model that
can adjust to such temporal changes in productivity (Perälä and Kuparinen, 2017; Tirronen et al., 2022).

In order to better predict future resource availability and adapt management accordingly, it is important to understand how
anthropogenic stressors, such as fishing and climate change, affect recruitment dynamics. In this study, we therefore focused
on the stock-recruitment relation, where we considered changes in spawner weight and sea temperature in addition to spawner

abundance for modeling the recruitment production in 17 Atlantic cod stocks. Atlantic cod is a commercially highly valuable
species, supporting a long-standing fishery. While the different cod stocks display an extraordinary high diversity in life-
history traits, geographical range and socio-economic context, each stock has experienced high rates of exploitation followed
by strong biomass declines (Frank et al., 2016; Lilly et al., 2008). Between the early 1960s and the early 1990s, the combined
spawning stock biomass of Northwest Atlantic cod stocks is estimated to have declined by more than 90 % (Hutchings and

Rangeley, 2011) and many stocks remain at unsustainable levels since then, despite reductions of directed fishing pressure
(e.g. (Hutchings, 2015a)). The different stocks are located in the North Atlantic Ocean (see Appendix A: Fig. A1), where
direction and intensity of the recruitment response to ocean warming depends on the stock's geographical position (Mantzouni
and Mackenzie, 2010; Planque and Fredou, 1999; Drinkwater, 2005; Brander, 2010). Changes in recruitment production have
been attributed to age-size truncation, but also Allee effects (e.g. (Marshall et al., 2006; Neuenhoff et al., 2018; Van Leeuwen

et al., 2008; Dean et al., 2019; Buren et al., 2014; Rose, 2004)), though the little empirical evidence has led to a general low
acceptance of Allee effects in Atlantic cod.



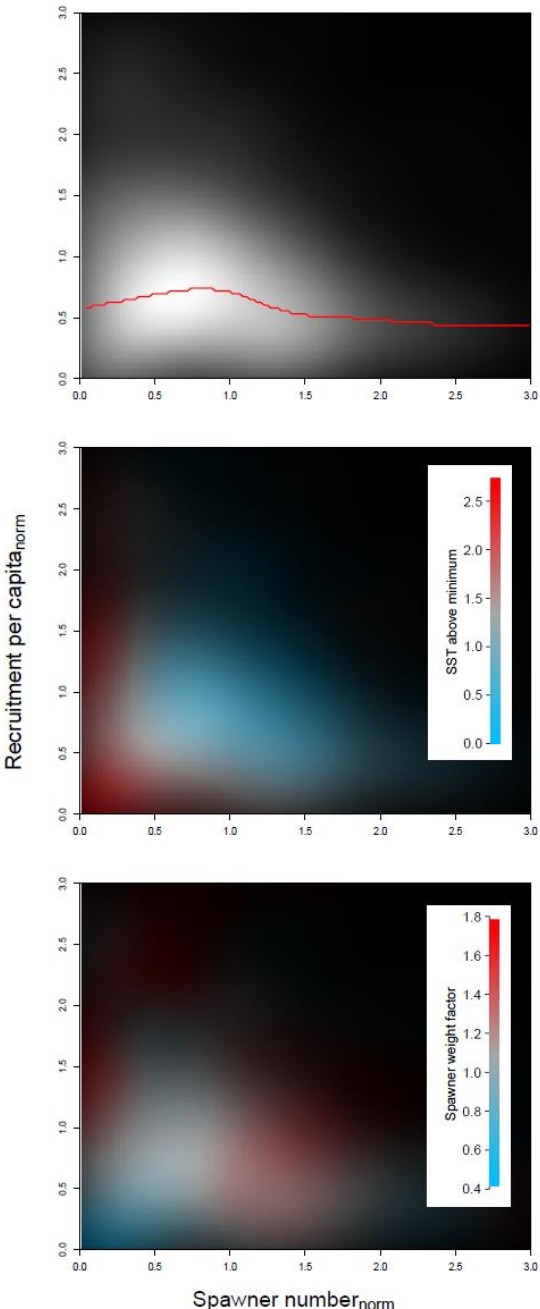

**Figure 1: Density plot of stock assessment data of normalized recruitment per capita ratios (y-axis) and normalized spawner abundance (x-axis). a) shows all data, with probability density shown as brightness, where red line shows most probable recruitment**



**per capita; colours in b) show in addition the sea surface temperature (SST) in comparison to average experienced SST (red indicates above average SST, blue indicates below average SST); and in c) colours show the spawner weight in comparison to average spawner weight (blue indicates below average weighing spawners, red indicates above average weighing spawners).**

A density-plot of all "raw" stock assessment data from all 17 cod stocks confirms the highest probability for a decrease in recruitment per capita ratios (recruitment / SSB) at below average spawner abundance, indicating patterns of an Allee effect

(Fig. 1 a). Interestingly, most stocks have been prevalent in the area of the Allee effect threshold (Fig. 1 a, white area), where SSB degradation can have strong repercussions for population and management. In particular at low spawner abundance, below the Allee effect threshold, recruitment per capita ratios are accompanied by temperatures above the average experienced sea surface temperature (SST) (Fig. 1 b, red shading), as well as a below average spawner weight (Fig. 1 c, blue shading). We therefore hypothesize that consideration of spawner abundance, spawner weight and SST as components of the stock-

recruitment function, should give insight on Allee effect dynamics in Atlantic cod. Our objective is the description of the data with an alternative stock-recruitment approach that can disentangle and quantify the impact of spawner abundance, weight and sea surface temperature on recruitment in order to reveal potential drivers of Allee effects. In contrast to many conventional stock-recruitment models, we do not use SSB as an aggregate, but consider heterogeneity in recruitment production among spawners of different weight. Though maternal effects have been included in models before, we here allow for stock-specific,

non-linear spawner weight, abundance as well as SST effects on recruitment. We link the stock-recruitment function to an age-structured population model, which is parameterized for each stock separately. While common stock-recruitment models consider the Allee effect as a static, density-dependent population property, this approach allows an Allee effect to also emerge or disappear from changes in spawner weight and sea temperature. Finally, we relate presence and strength of the Allee effects to stock recovery to discuss the role of Allee effects for rebuilding from depletion.

## 2. Material and Methods

### 2.1 Data

For the 17 Atlantic cod stocks, we extracted time series on recruitment, SSB, age-specific abundance, fishing mortality and life history traits (weight, probability of maturation and natural mortality) from publicly available assessment reports issued by the different fisheries institutions responsible (ICES (International Council for the Exploration of the Sea, www.ices.dk),

DFO (Department Fisheries and Oceans Canada, www.dfo-mpo.gc.ca), NAFO (Northwest Atlantic Fisheries Organization, www.nafo.int) and NOAA's Northeast Fisheries Science Center (www.nefsc.noaa.gov)). Stock specific number of age classes (5-13 classes), recruitment age (age 1-3) and the different length of time series available (17-68 years) was considered in the



model (Appendix A: Table A1). Gaps in the time series of life history traits were completed by taking the average or using data from different reports in order to keep the time-series as long as possible.

Time series of sea surface temperature, SST, from each stock's geographical location was extracted from NOAA's Earth System Research Laboratory, Physical Sciences Division (NOAA_ERSST_V4 data, www.esrl.noaa.gov/psd/). Average annual SST varies between around 3 °C in the northern North Atlantic Ocean where North East Arctic (NEA) and the Norwegian Coastal cod stock are located and around 15 °C in the southernmost area of Atlantic cod (Flemish Cap stock; Appendix A: Fig. A1). See Appendix B for a summary and description of the data and its sources used.

### 2.2 Population dynamics

We used an age-structured population model (Caswell, 2001) to describe the population dynamics of the different Atlantic Cod stocks. The model was parameterized for each stock separately and structured according to the stock's age classes $a \in [1, A]$, with $A$ the stock-specific maximum age class. Note that the first age class describes recruitment, while the age of recruitment $a_R$ varies among stocks (Appendix B: Table B1).

Recruitment production $N(y, 1)$ depends on the abundance of individuals in age class $a$ in spawning year $y_s$, $N(y_s, a)$, accounting for their age-specific probability of maturation $P(y_s, a)$ and their individual weight when spawning $W(y_s, a)$. Because for various Atlantic cod stocks, variation in recruitment production has been linked to changes in sea surface temperature, SST (Drinkwater, 2005; Planque and Fredou, 1999; Brander, 2010), we further considered recruitment production to be influenced by SST:

$$N(y, 1) = R(N(y_s, a), P(y_s, a), W(y_s, a), \text{SST}(y)),$$  (1)

the year of spawning is defined as $y_s = y - a_R$.

Except the just produced recruitment, each age class is exposed to age-specific fishing mortality $F(y, a)$ and natural mortality $M(y, a)$. Thus, the abundance of each class is:

$$N(y, a) = N(y - 1, a - 1)e^{-F(y-1, a-1) - M(y-1, a-1)}, a \in [2, A - 1].$$  (2)

The oldest age class $A$ is considered a plus group and accumulates all fish of the stock-specific maximum age class and older:

$$N(y, A) = N(y - 1, A)e^{-F(y-1, A) - M(y-1, A)} + N(y - 1, A - 1)e^{-F(y-1, A-1) - M(y-1, A-1)}.$$  (3)

### 2.3 Stock-recruitment function with separable effects

In contrast to the classical stock-recruitment models using spawning stock biomass, SSB, as an aggregate (Ricker, 1954; Beverton and Holt, 1957), abundance and weight of spawners were considered *separately* for recruitment production. Thus,

the per unit biomass fecundity and egg quality was not assumed to be equivalent among spawner of different weight. Further,



in contrast to models that do consider maternal effects (e.g. (Shelton et al., 2015; Marteinsdottir and Thorarinsson, 1998; Brunel, 2010)), stock-specific and nonlinear effects of spawner weight was considered. We did this by introducing a stock-recruitment function with three separable effects on the recruitment production in Atlantic cod, which are: 1) the basic demographic component, which aggregates the abundance of spawners (spnum) with their age-specific average weights; 2)

changes in average weight of spawners (spwe) and 3) sea surface temperature, SST. Without significantly increasing the data demand in comparison to most conventional stock-recruitment models, the introduced stock-recruitment function is able to capture difference in significance, strength and direction of either effect between stocks. It further shows the diversity in density dependences in the recruitment production of Atlantic cod stocks. Note that the introduced stock-recruitment model relies on time series of SSB for validation and the SSB data itself is a result of a stock-assessment model and its respective model

assumptions. Our approach offers therefore an alternative view, but is not intended to replace existing stock-assessment practice. It can be used in practice to simulate population dynamics scenarios and analyze projections to give management decision support.

The first factor, spnum, describes the effect of abundance changes of the spawning population on recruitment production. Spawners of average weight are mature individuals at their age-specific typical weight $\overline{W}$. $\overline{W}(a)$ is gained by taking for each

age class the average of the weight time series. SSB at average weight $SSB_N$ is given by:

$$SSB_N(y) = \sum_1^A N(y,a)P(y,a)\overline{W}(a). \tag{4}$$

The impact of spnum on recruitment production is only due to changes in abundance and/or probability of maturation. Because probability of maturation is not changing much over time, we are confident that spnum captures mainly the impact of spawner abundance.

The relation between recruitment production and $SSB_N$ is captured by a sigmoidal function of spawner abundance $H(SSB_N)$, where recruitment per capita can, dependent on the parametrization, show both compensatory and depensatory dynamics of different steepness, including predator-pit like Allee effects for recruitment per capita (Appendix C: Fig. C1). It can therefore capture the diversity in recruitment dynamics of the different Atlantic cod stocks.

$$H(SSB_N) = \frac{1}{1+exp(-k(SSB_N - SSB_0))}. \tag{5}$$

$SSB_0$ is the position of the inflection point, where the rate of recruitment production is highest, and $k$ is the function's steepness at $SSB_0$. See Appendix C: Fig. C1 for a schematic figure with all parameters explained.

The second factor, spwe, captures the effect of deviations in average spawner weight on recruitment production. Because $SSB_N$ is $SSB$ at average spawner weight, Eq. 6 depicts the year specific *deviations* from average spawner weight. It can be interpreted as a proxy for spawner fitness. While most conventional stock-recruitment model assume fecundity and egg production to

increase linearly with spawner weight (e.g. (Hilborn and Walters, 1992; Hutchings, 2005)), in this study we allow for a





nonlinear dependence. We do this by introducing the exponent $c$, which quantifies the effect deviations from average spawner weight have on recruitment production. While $c = 1$ shows cases of linearity, as in most conventional stock-recruitment models using SSB as an aggregate, but also many models considering maternal effects (e.g. (Brunel, 2010; Shelton et al., 2015; Marteinsdottir and Thorarinsson, 1998)), $c > 1$ means that increasing spawner weight leads to an unproportional strong

increase in recruitment production (and vice versa decreasing spawner weight leads to unproportional strong decrease in recruitment; convex down) and $c < 1$ implies an unproportional slow increase in recruitment production with increasing spawner weight (convex up). Appendix C: Fig. C1 shows the effect of parameter $c$ on the function.

$$F(y) = \left(\text{SSB}(y)/\text{SSB}_N(y)\right)^c. \tag{6}$$

The third component of the stock-recruitment function is ambient SST, which is considered by a Lorentz function, $G$ (Eq. 7).

Because recruitment response to SST anomalies is stock-specific, the bell-shaped Lorentz function was chosen in contrast to the conventional approach of modeling a monotonic temperature dependence effect (e.g. (Planque et al., 2003; Clark et al., 2003; Hilborn and Walters, 1992) though recent studies do recognize its time-variance (e.g. (Ottersen et al., 2013; Stige et al., 2013; Szuwalski et al., 2015; Olsen et al., 2011)). Laurel et al. (2017) recently applied a Lorentz function to describe growth dependence on SST and size in juvenile Arctic cod. A Gaussian function did not perform well, because some stocks show a

nearly linear response to SST increase. Parameter $b$ of the Lorentz function describes the width of the temperature optimum curve. See Appendix C: Fig. C1 for a description of each parameter. For five stocks, the Lorentz function finds a temperature optimum $T_0$, within the observed range of data (Appendix C: Fig. C2). For stocks where no temperature optimum in the range of observations was found, the whole temperature range in simulations localized on one side of the temperature curve (Appendix C: Table C1), e.g. in case of NEA and Faroe Plateau stocks (Appendix C: Fig. C2). Appendix C: Table C1

summarizes the temperature optima and width parameters of the fitted Lorentz function. The fitted temperature optima for recruitment production are within the documented range for Atlantic cod (e.g. (Planque and Fredou, 1999; Drinkwater, 2005; Righton et al., 2010; Pörtner et al., 2001)), though depending on the physiological and/or ecological mechanism considered. Depending on the current position on the temperature optimum curve (red mark, Appendix C: Fig. C2), changes in future SST have a different impact on recruitment production. We estimated temperature sensitivity (Table 1) as the ratio between

recruitment at 0.5 °C SST increase and recruitment at present SST. In Table 1 we show it as a percent change of projected recruitment from present SST.

$$G(\text{SST}) = \frac{1}{1+b^2(\text{SST}-\text{SST}_0)^2}. \tag{7}$$

Considering the three effects on recruitment production, spnum, spwe and SST, the stock-recruitment function capturing all different 17 Atlantic cod stocks, can thus be formulated as:





$N(y, 1) = R(y) = L' \, H\big(\mathrm{SSB}_N(y_s)\big) \, F(y_s) \, G\big(\mathrm{SST}(y)\big).$ (8)

where $y_s$ is the year of spawning. $L' = L \, \overline{\mathrm{SSB}} \, H(\overline{\mathrm{SSB}}) \, G(\overline{\mathrm{SSB}})$, where $L$ is a normalization parameter needed to reduce the scale of the different cod stocks. $L'$ is the recruitment production at stock-specific average SSB, average SST and average spawner weight. The stock-specific coefficients $L$, $k$, $c$, $b$, $T_0$ were fitted by an optimization procedure, using time series of SSB (see below). The exact values of the coefficients are given in Appendix C: Table C1.

**2.4 Definition of the Allee effect strength**

The Allee effect is defined as a decline in the individual growth rate or recruitment per capita ratio, $R_{\mathrm{cap}}$, at small population density or abundance (e.g. (Courchamp et al., 1999; Hutchings, 2015a)). The threshold below which recruitment per capita decreases is termed the Allee effect threshold. If the individual growth rate decreases to zero at a positive population abundance, the Allee effect is usually considered strong (Berec et al., 2007; Hutchings, 2015a). Here, we estimated the Allee

effect strength, $A_s$, as the ratio between $R_{\mathrm{cap}}$ at the Allee effect threshold $\mathrm{SSB}_0$ and the minimum $R_{\mathrm{cap}}$ at $\mathrm{SSB}_N$ below the Allee effect threshold (Eq. 9). Without an Allee effect, $\mathrm{SSB}_0$ is simply the inflection point. We considered $\mathrm{SSB}_N$ in order to focus on spawner abundance, because Allee effects are density or abundance dependent phenomena. At the Allee effect threshold, $R_{\mathrm{cap}}$ is maximal and the corresponding biomass is $\mathrm{SSB}_0$. Because all stocks show to some extent a decline in the per capita growth rate (Fig. 3), an Allee effect can be quantified for each stock as the decline in recruitment per capita relative to its maximum

so that $A_s \in [0, 1]$. $A_s = 0$ when there is no Allee effect and $A_s = 1$ when recruitment per capita declines to zero. Appendix C: Fig. C1 shows a schema of the definition.

$A_S = 1 - \dfrac{\min(R_{cap}|\mathrm{SSB}_N < \mathrm{SSB}_0)}{\max(R_{cap}|\mathrm{SSB}_N > \mathrm{SSB}_0)}.$ (9)

**2.5 Optimization and model validation**

The stock-specific coefficients $L$, $k$, $\mathrm{SSB}_0$, $c$, $b$ and $T_0$ were found by an optimization procedure, using time series of SSB. The

optimization was accomplished by maximizing a logarithmic likelihood function (see Appendix D, Eq. D1) with an algorithm comprised of a sequence of Latin hypercube, Monte-Carlo and Nelder-Mead methods, which is common practice in stock assessment models (Nielsen and Berg, 2014; Cadigan, 2015). Model performance with the different stock-recruitment effects was compared with model fitting errors (Root Mean Squared Logarithmic Error, RMSLE) and the Akaike information criterion (AKAIKE; Appendix D: Table D1). We also compared simulated recruitment data with the recruitment and total biomass time

series from stock-assessments (Appendix D: Fig. D2-4). Overall, simulations are well within the range of the recruitment data





and documented confidence intervals. For details on the optimization procedure and validation of the model fit see Appendix D.

Model building, optimization and analysis was performed in R (R Core Team, 2017).

## 3. Results

### 3.1 SSB response to exploitation and SST

For all stocks, SSB simulated with the new stock-recruitment function follows the data trend and remains well within the documented confidence intervals of the data (Fig. 2). A high goodness of fit for total biomass (TB) and recruitment (R) time series is shown in Appendix D: Fig. D2-4). Within the last 25 years, all stocks experienced strong declines in SSB, with Northwest Atlantic stocks (Fig. 2, stars) having experienced an overall steeper decline than stocks in the Northeast Atlantic.

SSB follows the general patterns of fishing pressure with a minor lag (Fig. 2), which is expected given that both time series stem from the same stock assessment model. Fishing pressure increased stronger in the Northwest Atlantic stocks than in the Northeast Atlantic stocks and there is a general decreasing trend in fishing pressure in recent years (Appendix B: Fig. B2 c, d). Most stocks are now either on an upward or on a rather constant SSB trend following a stock decline, though a few stocks (Norwegian Coastal, Southern Gulf of St. Lawrence, Western Baltic and Gulf of Maine stock) are still declining (Fig. 2).

Average ambient sea surface temperature ranges between 3 °C and 11 °C (Appendix A: Fig. A2) and shows a significant (p = 0.0001) increasing trend (Appendix B: Fig. A1 i) with an average increase of 0.1°C for the last 20 years (1998-2018). On individual stock level, SST shows less consistent trends and the relation between SST and SSB is less clear. On a recruitment level (Appendix C: Fig. C2), 13 out of 17 stocks were shown to be effected by the dynamics of SST. Among these, only for the Southern Grand Bank and Celtic Sea stock, SST is identified as a stringent positive factor, while for six stocks (NEA,

Faroe Plateau, Gulf of Maine, North Sea, West of Scotland, Irish Sea stock), SST increase results in lower recruitment production (Appendix C: Fig. C2) and for another five stocks (Icelandic, Southern Gulf of St. Lawrence, Western Baltic, Kattegat, Georges Bank stock) SST impact is ambivalent and depending on the current temperature (Appendix C: Fig. C2). For the remaining stocks (Norwegian Coastal, Northern Gulf of St. Lawrence, Northern and Flemish Cap stock), the model does not reveal a substantial effect of SST in recruitment production.

The SST effect on SSB level is less clear and difficult to disentangle from exploitation effects (Fig. 2). For example, for the Northeast Arctic cod stock, recruitment response to SST is negative, but because current SST is at the lower end of the curve (Appendix C: Fig. C2) and fishing pressure is currently low, the SSB increase coincides with recent warming (Fig. 2). Similar



patterns are observable for the Icelandic and North Sea stock, which coincides with other studies (Kjesbu et al., 2014; Brander, 2010; Brander, 2018).

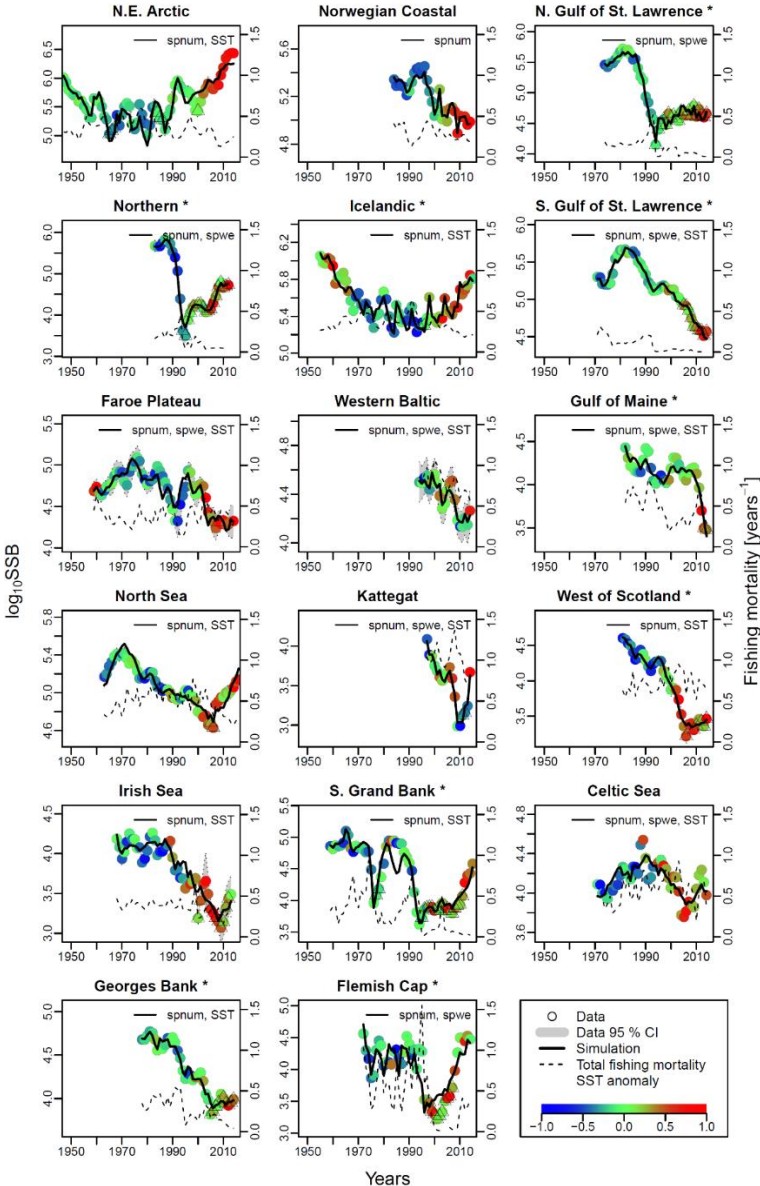


**Figure 2: Time series of spawning stock biomass (SSB), total fishing pressure and sea surface temperature SST for each stock. Stars mark Northwest Atlantic cod stocks, while the other cod stocks are found in the Northeast Atlantic. Dots show the stock assessment**





**data for SSB with dot colouring according to the matching year's average SST. Triangles depict years of collapse (SSB < 20 % of SSB$_{max}$ and fishing pressure > average fishing pressure). Where available, the 95 % confidence interval of the stock assessment data**
**is shown in grey. Solid lines indicate estimates based on the respective fit to data for the stock-recruitment function $H(SSB_N)$. Number of spawners at average weight (spnum) is considered as the basis, in addition to divergence of the average spawner weight (spwe) and/or changes in sea surface temperature (SST). The broken line shows the total fishing mortality.**

## 3.2 Recruitment production

The introduced new stock-recruitment function considers three factors influencing Atlantic cod recruitment production: 1) number of spawners at average weight (spnum), 2) deviation from the average spawner weight (spwe) and 3) changes in sea surface temperature (SST), where spwe and SST are considered as environmental factors. Spnum is the basic model and we tested whether consideration of in addition spwe and/or SST improves the model fit (i.e. to fit SSB stock assessment data). We find that in the majority of Atlantic cod stocks (13 stocks), consideration of SST in addition to spnum significantly improves

the fit to data (Appendix D: Table D1). For nine stocks, consideration of changes in average spawner weight improves the fit (Appendix D: Table D1). In six stocks, consideration of spwe and SST is significant (Appendix D: Table D1). Only in the Norwegian Coastal cod stock, spwe and SST do not improve the fit and (average weighing) SSB is best in describing data (Fig. 3: red broken line and black dots are the same). Remaining deviations between SSB data and simulated SSB are due to other, here not investigated, factors.





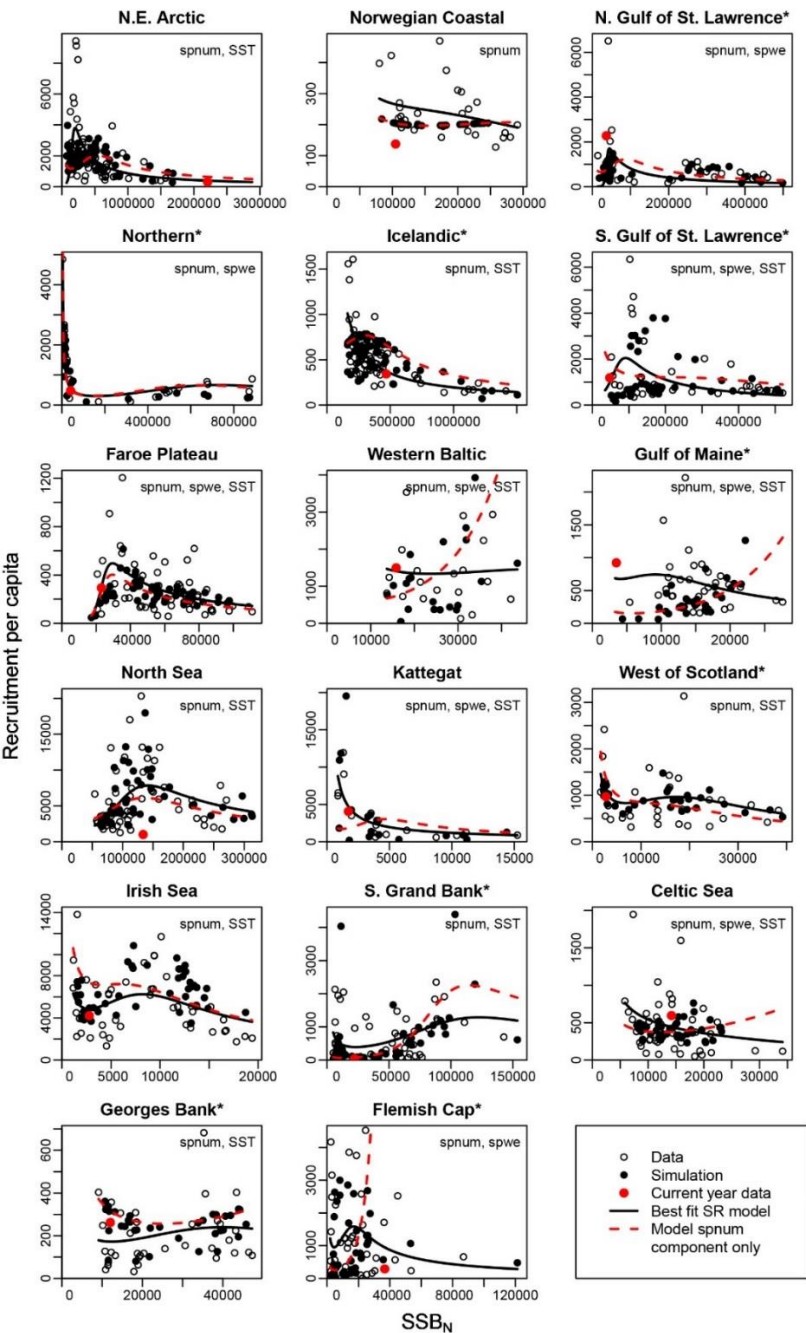




**Figure 3: Recruitment per capita as a function of spawner abundance at average weight, SSB$_N$. Circles show the stock assessment data, black dots show the simulated results of the respective best model. The black line shows the function $H(\text{SSB}_N)$ direct fit to data. The red dots indicate the most recent stock assessment data. The dashed red line indicates the relation between recruitment per capita and SSB$_N$ considering only number of spawners at average weight (spnum) and neglecting effects of spawner weight (spwe)**
**and sea surface temperature (SST). Thus, deviations between the solid and broken lines is explained by the impact of spawner weight and/or SST.**

Figure 3 shows the simulated recruitment per capita ratios. The red broken line shows the spawner abundance component of the stock-recruitment function, spnum, which shows changes in the recruitment per capita ratios due to the spawner abundance

factor. We consider this the *inherent* density-dependence regulation. If recruitment per capita ratios decline at low spawner abundance below a certain threshold, the Allee effect threshold, this can be caused by an abundance related, *inherent* Allee effect. For 13 out of 17 stocks, we find an inherent Allee effect, though with highly varying Allee effect strength ($A_s$, Table 1) and locations of the Allee effect threshold (SSB$_0$, Table 1). For a few stocks, the model suggests extremely high recruitment per capita ratios and the Allee effect threshold outside the observational range, which stands in contrast to conventional

threshold localizations at low abundance (Stephens et al., 1999; Courchamp et al., 1999; Hutchings, 2015a). Our Allee effect strength definition (Eq. 9) is further challenged by stocks that show predator-pit shaped recruitment per capita relations (predator driven Allee effect), where recruitment per capita drops at low abundance ("pit area"), but increases again at very low and high abundance (Gascoigne and Lipcius, 2004; Swain and Benoit, 2015). The model confirms the persistent predator-pit shaped curve for the Northern cod stock (Swain and Benoit, 2015; Shelton and Healey, 1999), but also finds similar, but

weaker pattern for the West of Scotland, Irish Sea and Southern Grand Bank stock.

If recruitment per capita ratios decline at low abundance due to changes in the average spawner weight and/or SST or changes in the average spawner weight and/or SST enhance the decline of recruitment per capita ratios, i.e. in the best model compared to its spnum component, we consider this an *emergent* Allee effect. Note that we investigate the significance of SST *deviation* from the average ambient SST and, similarly, we investigate significance of spawner weight *deviation* from the average

weighing SSB. The majority of stocks show an emergent Allee effect. In six stocks, the inherent Allee effect is strengthened (Table 1). For example, in the Northeast Arctic stock, an emergent Allee effect is attributed to SST changes, while in the Northern Gulf of St. Lawrence stock changes in spawner weight strengthen an Allee effect (Fig. 3, Table 1). Weaker Allee effect examples include the North Sea cod, for which we find SST to strengthen the inherent Allee effect, which confirms results by Winter et al. (2020). In the Southern Gulf of St. Lawrence, West of Scotland and Irish Sea cod environmental factors

(spwe and/or SST) induce an Allee effect that was not there before.

Significance of spawner weight changes for Allee effect strength is confirmed by the exponent parameter $c$, which is $> 0$ for the majority of stocks (Table 1). And in most of these stocks, recruitment production strongly ($c > 1$) increases with increasing





spawner weight or, in other words, production declines strongly with decreasing spawner weight. Thus, for stocks with an inherent Allee effect, a decline in spawner weight strengthens the Allee effect. Strengthening of the inherent Allee effect in

the Faroe Plateau stock is therefore mainly due to SST changes with a relatively weak spwe effect ($c = 0.7$, Table 1).

For stocks where SST was found to be significant for the Allee effect (e.g. NEA, Southern Gulf of St. Lawrence and North Sea cod stock), appearance of the Allee effect is indeed also accompanied by strong changes in SST (cooling for NEA cod, warming for Southern Gulf of St. Lawrence and North Sea cod stocks, Fig. 2). The NEA cod stock was in addition much more sensitive to SST changes during the cooling period than at present (Appendix C: Fig. C2). Ocean warming probably helped

the stock to recover from low abundance and the Allee effect region.

In a few stocks, we find compensatory behaviour, the opposite of an Allee effect, to be strengthened when considering changes in spawner weight and/or SST. In five stocks (Icelandic, Western Baltic, Gulf of Maine, Kattegat, Celtic Sea stock) this leads to disappearance or masking of the inherent Allee effect and in two stocks (Southern Grand Bank and Flemish Cap stock), the inherent Allee effect is weakened (Fig. 3, Table 1).


**Table 1: Fitted stock-recruitment model estimates and Allee effect characteristics for all Atlantic cod stocks.** $c$ **describes the strength of linearity between spawner weight change and recruitment production, sensitivity to SST describes the change in recruitment per capita with 0.5 °C based on current ambient SST. $SSB_0$ is the Allee effect threshold (or the inflection point if no Allee effect is present) and $A_s$ is the Allee effect strength, which we differentiate between only abundance related (inherent) and related to SST and/or**

**spawner weight changes (emergent). The recovery time is estimated as the number of years where SSB is below 20 % of maximum SSB and fishing pressure is below average fishing pressure.**

| Atlantic cod stock name | $c$ | sensitivity to SST [% / 0.5 °C] | $SSB_0$ in % of $SSB_{MAX}$ (inherent) | $SSB_0$ in % of $SSB_{MAX}$ (emergent) | $A_s$ inherent | $A_s$ emergent | recovery time [years] | change in Allee effect (A.E.) |
|---|---|---|---|---|---|---|---|---|
| N.E. Arctic | no W sensitivity | -40 | 20 | 8 | 0.42 | 0.82 | 3 | SST enhances A.E. |
| Norwegian Coastal | no W sensitivity | no SST sensitivity | 85 | 42 | 0.06 (no A.E.) | 0 (no A.E.) | no collapse | No A.E. |
| N. Gulf of St. Lawrence | 3.3 | no SST sensitivity | 13 | 8 | 0.47 | 1 | 21 | Spwe enhances A.E. |
| Northern | 3.2 | no SST sensitivity | 75 | 86 | 0.53 | 0.55 | 13 | No A.E. change |
| Icelandic | no W sensitivity | -30 | 23 | outside of range | 0.11 | 0 (no A.E.) | 1 | No A.E. |



| | | | | | | | |
|---|---|---|---|---|---|---|---|
| S. Gulf of St. Lawrence | 2.2 | -39 | 44 | 15 | 0 (no A.E.) | 0.71 | 10 | A.E. emerges |
| Faroe Plateau | 0.7 | -25 | 23 | 23 | 0.84 | 0.88 | 1 | SST enhances A.E. |
| Western Baltic | 6 | -48 | outside of range | outside of range | 0.89 | 0.05 (no A.E.) | no collapse | A.E. threshold outside obs. range |
| Gulf of Maine | 3.7 | -23 | outside of range | 28 | 0.88 | 0.09 (no A.E.) | 2 | A.E. threshold outside obs. range) |
| North Sea | no W sensitivity | -26 | 39 | 41 | 0.46 | 0.77 | 2 | SST enhances A.E. |
| Kattegat | 3.3 | -48 | 39 | 4 | 0.43 | 0 (no A.E.) | 2 | Spwe and SST mask A.E. |
| West of Scotland | no W sensitivity | -24 | 28 | 36 | 0 (no A.E.) | 0.14 | 3 | A.E. emerges |
| Irish Sea | no W sensitivity | -38 | 31 | 37 | 0.01 (no A.E.) | 0.22 | 8 | A.E. emerges |
| S. Grand Bank | no W sensitivity | 276 | 100 | outside of range | 0.97 | 0.7 | 8 | SST weakens A.E |
| Celtic Sea | 4.4 | 20 | outside of range | 22 | 0.46 | 0 (no A.E.) | no collapse | A.E. threshold outside obs. range |
| Georges Bank | no W sensitivity | -48 | outside of range | 54 | 0.22 | 0.28 | 5 | A.E. threshold outside obs. range |
| Flemish Cap | 5.2 | no SST sensitivity | outside of range | 37 | 1 | 0.4 | 10 | A.E. threshold |



| | | | | | | | | outside obs. range |
|---|---|---|---|---|---|---|---|---|
| | | | | | | | | |

## 3.3 Relation between Allee effect, depletion and recovery

We find that in several Atlantic cod stocks, SSB is below the stock-specific Allee effect threshold ($SSB_0$, Table 1), where a

rise in recruitment per capita ratios is constrained by an Allee effect (e.g. Northern, Southern Gulf of St. Lawrence, Faroe Plateau, North Sea, Southern Grand Bank stock) (red dot, Fig. 2). Because for most stocks, SST is significant, but the SST impact on recruitment critically depends on the position on the Lorentz curve (Appendix C: Fig. C2), we estimated a temperature sensitivity, which is the projected recruitment per capita change with 0.5 °C (Table 1). With exception of the Southern Grand Bank and Celtic Sea stock, overall future recruitment response to rising SST is projected to be negative with

recruitment of the Western Baltic, Kattegat and Georges Bank stocks projected to be reduced up to 50 % (Table 1). Thus, in the Southern Gulf of St. Lawrence, Faroe Plateau and North Sea stock, which are currently degraded and in addition show an Allee effect that is strengthened by SST, recovery is likely increasingly hampered by future ocean warming.

In Figure 4, we related depletion level to recovery time and Allee effect strength in order to see whether the Allee effect influences recovery. We estimated stock recovery time as the longest time span of consecutive years where SSB remained

below 20 % of its maximum, $SSB_{max}$, and while fishing pressure was lower than average. Stock recovery time ranges between 1 years (Faroe Plateau stock; fishing pressure was increased again after a year though SSB remained below 20 % of $SSB_{max}$) and 21 years (Northern Gulf of St. Lawrence stock). In accordance with other studies (Hilborn et al., 2014; Neubauer et al., 2013; Hutchings, 2015a), we find that the stronger the severity of depletion, the longer it takes the stock to recover. We do not find a correlation between Allee effect strength (red colour, Fig. 4) and recovery time, though on average stocks with an Allee

effect show a longer recovery time. All stocks with an Allee effect did collapse (fell below 20 % $SSB_{max}$) and fell below their emergent Allee effect threshold ($SSB_0$, Table 1). An exception is the Western Baltic cod, where the model estimated an Allee effect threshold outside the observational range (Fig. 3). Thus, magnitude of degradation and presence of an Allee effect matter, but not Allee effect strength or the location of the Allee effect threshold.



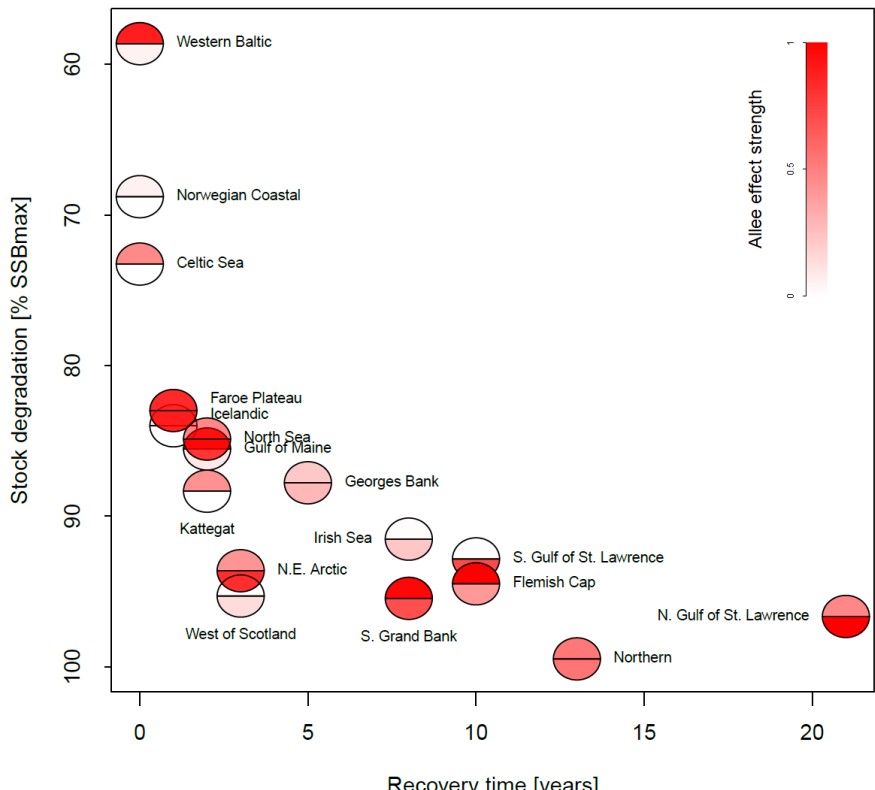

**Figure 4: Association between the maximum depletion level and recovery time for all stocks. Maximum depletion level is the ratio between minimum SSB and 20 % of SSB$_{max}$, the considered collapse threshold level. Recovery time was estimated as years of SSB < 20 % SSB$_{max}$ and fishing pressure below average level. Colour intensity of the lower semicircle corresponds to the emergent Allee effect strength and colour intensity of the upper semicircle to the inherent Allee effect strength.**

## 4. Discussion

### 4.1 Presence of Allee effects in Atlantic cod

Slow or negligible recovery in several marine fish stocks and in particular in Atlantic cod stocks remains a challenge. In this study, spawner abundance, changes in spawner weight and temperature were considered as drivers of Atlantic cod recruitment production in order to link these with population trajectories. As a potential impediment for recovery, this study in particular looked at evidence and causes for Allee effects in Atlantic cod. Identification of potential drivers of Allee effects aid in finding (precautionary) fisheries management strategies that can facilitate recovery of depleted stocks.





We find for the majority of Atlantic cod stocks a depression of the per capita growth rate at low to intermediate abundance, which, strictly speaking, could be considered an Allee effect, even if the depression is small. The Allee effect shows a high variety in its form (steep and flat curves, predator-pit shape), location of the Allee effect threshold (8 % - 100 % of $SSB_{max}$) and Allee effect strength $(0 - 1)$ (Table 1). Our results confirm the ongoing discussion of potential Allee effects in Atlantic

cod, which so far has only focused on single stock studies (e.g. (Cabral et al., 2013; Kuparinen and Hutchings, 2014a; Kuparinen et al., 2014; Perälä et al., 2022)) or meta-analysis with aggregated data (e.g. (Keith and Hutchings, 2012; Myers et al., 1995; Hilborn et al., 2014)). To our knowledge, this is the first study showing and investigating the potential of Allee effect dynamics for every individual stock.

For several stocks, we find an inherent Allee effect (Fig. 3, Table 1) that is regulated by the abundance of spawners, which has

been suggested by others. Some examples include (mechanism in brackets): NEA (North East Arctic; a sex bias leading to a reduction in total egg (Marshall et al., 2006)), Gulf of St. Lawrence (increased predation by seals at low abundance (Neuenhoff et al., 2018)), Northern cod (increase in spawner mortality (Kuparinen and Hutchings, 2014b) and disruption of sub-population structure (Frank and Brickman, 2000)), Baltic cod (low food availability after collapse (Van Leeuwen et al., 2008)) and Gulf of Maine (shrinking sub-population (Dean et al., 2019)). For several stocks, the model finds very large recruitment per capita

ratios and Allee effect thresholds that are either outside the observational range or at high spawner abundance (e.g. Southern Grand Bank cod and emergent Allee effect in Northern cod). These cases challenge our definition of an Allee effect, which is usually confined to low abundance (e.g. (Stephens et al., 1999; Courchamp et al., 1999)). We argue here that recruitment dynamics might display a large variety of density-dependences, even within the same species, and that recruitment per capita ratios can show a positive density-dependence at larger abundances than normally assumed. A possible reason could be that

the data used here is only a snapshot of already highly degraded stocks. A prominent example is the Northern cod, which seems to have an Allee effect threshold at 75-86 % of its maximum observed SSB (Table 1), though we know that the stock has been a multiple of its biomass before its collapse in the 1990's (Dfo, 2019). Thus, the time-series we have available may not be a good representation of the stock's complete SSB history. Longer time-series could aid in better locating Allee effect thresholds, in particular those located at large SSB.

Besides spawner abundance influencing the occurrence of Allee effects, we here identified changes in spawner weight as well as changes in sea surface temperature as additional, environmental factors affecting the presence of Allee effects. The de-facto visible Allee effect we call here *emergent* Allee effect. SST change strengthens in particular the inherent Allee effect of the NEA, Southern Gulf of St. Lawrence and the North Sea cod; the latter has been recently shown to be impacted by a strong Allee effect strengthened by temperature rise (Winter et al., 2020). Note that the inherent Allee effect of the NEA cod is

strengthened during an exceptionally cool period, while the inherent Allee effect in the North Sea cod is strengthened during





a warm period (Fig. 2). Vice versa, SST changes can also counteract an inherent Allee effect, when positively affecting recruitment production. The inherent Allee effect of the Western Baltic, Gulf of Maine, Kattegat, Southern Grand Bank and Celtic stock was weakened or disappeared due to the change in SST. One suggested explanation is that the stocks were in particular sensitive to below average sea temperatures (strong, positive temperature dependence of the recruitment production)
when they were at low abundance (Fig. 2, Appendix C: Fig. C2). Inherent Allee effects of the Western Baltic, Gulf of Maine, Kattegat, Celtic Sea and Flemish Cap stock was reduced due to changes in spawner weight, though only for the Kattegat and Flemish Cap stock we find clear evidence for an increase in spawner weight in recent years (Appendix E: Fig. E1).

In contrast, changes in spawner weight strengthened the inherent Allee effects in the Northern Gulf of St. Lawrence stock ($c$ = 3.3, Table 1) and helped cause Allee effect appearance in the Southern Gulf of St. Lawrence stock ($c$ = 2.2, Table 1), where
indeed also a shift towards lighter spawners can be seen (Appendix E: Fig. E1). The stocks also shows a decrease in maturation age and spawner abundance in recent years (Appendix E: Fig. E2-E3), which has been linked to increased natural mortality and decreased productivity (Swain, 2011).

While we may be able to point out which factor is responsible for influencing presence of an Allee effect, it remains open which exact mechanism is responsible. As purely abundance (density) related mechanisms, for example a reduced chance and
success of mate finding, egg fertilization, group-protection, group-learning and less spawning migration at low abundance (Rowe et al., 2004; Rowe and Hutchings, 2003) has been proposed (and see stock-specific mechanisms mentioned above). Because of the positive relation between spawner mass, fertility and recruitment success (Marteinsdottir and Steinarsson, 1998; Hixon et al., 2013), changes in spawner weight can facilitate an Allee effect when the spawner population shifts towards lighter, smaller spawners. Such a shift can be induced by a change in body growth (e.g. induced by temperature rise), but is
also often induced by heavy fishing targeting the largest and most reproductive individuals, albeit *evolutionary* shifts towards younger spawners could also imply an increased reproductive output counteracting an Allee effect. This could explain why we do not find consistent evidence for a decrease in spawner weight though findings of an Allee effect *strengthening* by changes in spawner weight. Indeed, life history traits, such as the probability of maturation at different ages, is for most Atlantic cod stocks highly dynamic (Appendix E: Fig. E3), but we cannot distinguish between phenotypic and evolutionary changes in the
(spawner) population.

Changes in SST can strengthen the Allee effect by, for example, decreasing recruitment production, affecting food availability for spawners or recruitment, increased mortality of spawners or recruitment and affecting spawner growth. Note that we model the impact of spawner weight and SST independent of spawner abundance and density. Changes in spawner weight and sea temperature mainly strengthen (or weaken) the inherent Allee effect, which is already present (Table 1). The term "emergent"
may therefore be better replaced by "apparent", though we wish to draw with this term attention to the potential of Allee effects



to *emerge* due to changing population or environmental conditions. While density-independent depression of the per capita growth rate strengthens an already existing Allee effect, for Allee effect *emergence*, depression of the recruitment per capita growth rate needs to be density-dependent.

It is challenging to differentiate between emergent Allee effect and the coincidental appearance of unfavourable conditions
while the stock happened to be at low abundance if there is no data of unfavourable conditions also at high abundance. While the first is a real Allee effect, becoming effective as soon as the stock's abundance drops below a certain threshold, the second only *appears* as an Allee effect under conjunction of certain circumstances, but is not triggered by or dependent on stock abundance. This is important for predictions under future spawner and environmental changes because the stock may respond differently.

We consider that SST and spawner weight are not independent, so the effect of SST on spawner productivity is to some extent included in the spawner weight factor, for example, via food availability. Therefore, the modelled SST effect on recruitment per capita is the effect on recruitment itself (not via spawners) which can be as well a mechanism related to larval feeding-dependent survival to recruitment. When it comes to the relation between food (plankton) and cod survival, food abundance, food size (quality) and encounter rate are the main important factors influencing (Beaugrand et al., 2006). This question could
be further addressed by incorporating data on zooplankton abundance in recruitment feeding areas or phytoplankton-zooplankton–larvae–recruitment models, as for example for NEA cod (Endo et al., 2022). Another possible explanation of correlation between SST and larval survival to recruitment can be changes in circulation patterns which effect both SST and larvae drift. This mechanism could be analyzed by integrating the developed model with existing detailed larvae drift and ocean circulation data and models, as for example estimated for Georges Bank cod (Lough et al., 2006). If one adds any more
mechanisms behind the SST changes (e.g., through oceanographic features), that would increase the necessary spatial (e.g. not only horizontally, also vertically in the water column) and time resolution (e.g. seasonality) of the data. So, it remains to be tested whether a stock-recruitment model as in our paper would be sufficient to capture those factors or individual based models would be needed.

## 4.2 When can prolonged recovery be linked to an Allee effect?

We identified years of recovery, when SSB remained below the collapse threshold (SSB < 20 % $SSB_{max}$) even though fishing was reduced below its historic average. For many stocks, we find that in periods of prolonged recovery (triangles, Fig. 2), biomass levels are also below the Allee effect threshold. We also find that in many cases, fishing reductions were only in place long after the stock had fallen below its Allee effect threshold (e.g. Northern, North Sea, West of Scotland, Southern Grand Bank, Georges Bank and Flemish Cup cod), an observation shared by others (Neubauer et al., 2013). This is surprising, given



that the precautionary and biomass limit reference points used in fisheries management are often higher than the emergent
Allee effect thresholds discovered here.

We here attempted a gradual quantification of the Allee effect strength, defined by the decline of the recruitment per capita at
low abundance (Eq. 9). We do not find a strong relation between inherent or emergent Allee effect strength and length of
recovery (Fig. 4). One reason could be that our Allee effect strength definition does not consider the presence of hysteresis,

which is caused when per capita growth rate drops to zero at positive abundance (Jiang and Shi, 2010). Decisive for recovery
is whether an emergent Allee effect is present and how strong the stock is degraded (Fig. 4). This implies that for conservation
measures, the strength and threshold of the Allee effect should be both considered in order to ensure that stocks with an Allee
effect still recover after depletion. The Allee effect threshold could provide guidance for locating precautionary reference
levels and the Allee effect strength could provide guidance for the type of measures implemented below the precautionary

level.

## 4.3 Prediction to future changes

Climate change will continue to cause a considerably warmer Atlantic Ocean (Ipcc, 2019) and our results show that increasing
SST will have negative repercussions for most of the Atlantic cod stocks that have already passed their thermal optimum. This
could push stocks with an emergent, SST related Allee effect further below their Allee effect threshold, squandering any

rebuilding efforts so far. While in the past, some stocks were positively affected by an SST change and thus could withstand
an inherent Allee effect, this positive SST effect can diminish under future conditions. For example, the inherent Allee effect
of the Kattegat cod stock was suspended by extraordinary low SST in the past, but future warming is likely to negatively
impact recruitment production, which could unmask the inherent Allee effect. Interestingly, we find for many stocks that the
location of the Allee effect threshold of the inherent and emergent Allee effect differ (Table 1). Thus, changing environmental

conditions could induce an Allee effect at unexpected SSB levels. In some stocks, such as the Southern Grand Bank cod, we
find increasing SST to be beneficial, which could further enhance SSB increase and push it above the Allee effect threshold.
The increasing evidence for (dynamic) Allee effects challenges the default assumption regarding compensatory recruitment
dynamic in fisheries stock assessments and calls for precautionary and adaptive management strategies, which involve regular
re-assessments of management reference levels. In particular, when the interaction of the inherent Allee effect with

environmental factors leads to low or no detectability of an Allee effect, this poses a high management cost when
environmental conditions become less favourable for recruitment production and cannot counteract an (inherent) Allee effect
anymore.





For this study, we introduced a different stock-recruitment function that modelled recruitment production as a three separable sub-functions of spawner abundance, SST and spawner weight. This way, stock trajectories similar to these assessments could

be simulated and the impact of spawner abundance, weight and sea surface temperature on recruitment dynamics could be disentangled and quantified separately, which is helpful in finding Allee effect mechanisms and predicting environmental interactions of the Allee effect. The suggested functions and standardization procedure allowed modelling the diversity in recruitment dynamics of all 17 Atlantic cod stocks and within species comparison of model parameters between the different stocks.

Migratory behaviour and sub-population dynamics influence the presence and detection of an Allee effect (Frank and Brickman, 2000), but could not be considered here, because we used stock-assessment data without spatial information. We therefore could also not consider spatial factors that can influence recruitment variability, such as e.g. shift of eggs and larvae from their advantageous habitat due to wind and ocean currents (aberrant drift) or direct spatial constraints on eggs and larvae (vagrancy) (Sinclair and Iles, 1989). We found the relations between recruitment and spawner weight, abundance and SST by

statistical fitting, but did not assume further mechanisms behind, in particular for the pre-recruitment stage. For example, a decline in recruitment production at low abundance and spawner weight could also be due to egg and larval losses from physical processes. Such a mechanism could be true for the Baltic cod, where smaller female spawners have been found to produce lower egg buoyancy, leading to a lowered reproductive success in oxygen depleted waters (Hinrichsen et al., 2016; Vallin and Nissling, 2000). In some cases our model predicts depensatory dynamics at high abundance, with a threshold above

historical range. Whether these stocks indeed show an Allee effect is debatable, as the Allee effect is usually considered a low abundance phenomena. More data at low abundance, spatial information and stock-assessment independent data would give more indication. Because our study heavily relies on the data from stock-assessment models, our conclusions can only be seen in light of these data and models.

## 5. Conclusions

In conclusion, our approach on modeling recruitment production shows some limitations and is not designed to replace common stock-assessment models, but provides new perspectives on non-linear recruitment dynamics in marine fish. With this study we contribute to the emerging recognition of dynamic Allee effects (Tirronen et al., 2022) that can interact with the environment (e.g. (Berec, 2019; Vet et al., 2020; Winter et al., 2020)). We find that Allee effects are common in Atlantic cod and are highly dynamic under different spawner weight and SST conditions. If present, the Allee effect hinders recovery. Our

findings advocate for the application of more precautionary management measures with lower fishing opportunities to





counteract the high uncertainties in Atlantic cod recruitment dynamics, which can occur at larger population size than previously thought.

**Appendix**

The appendix consists of 5 appendices describing in detail 1) the location of the different Atlantic cod stocks (A), 2) stock
assessment data and parameters used (B), 3) The different stock-recruitment function components (C), 4) the optimization procedure and model validation (D) and 5) observed changes in population characteristics according to stock-assessment data (E).

**Appendix A: Location of different Atlantic cod stocks**

For each Atlantic cod stock, time series of sea surface temperature, SST, from each stock's geographical location was extracted
from NOAA's Earth System Research Laboratory, Physical Sciences Division (NOAA_ERSST_V4 data, www.esrl.noaa.gov/psd/). Average annual SST varies between around 3 °C in the northern North Atlantic Ocean where NEA and the Coastal cod stock are located and around 15 °C in the southernmost area of Atlantic cod (Flemish Cap stock). Figure A1 shows the different Atlantic cod stocks located in the North Atlantic Ocean and with their average (from the time series used), ambient sea surface temperature. The different stocks show four distinct temperature regimes (Figure A2).



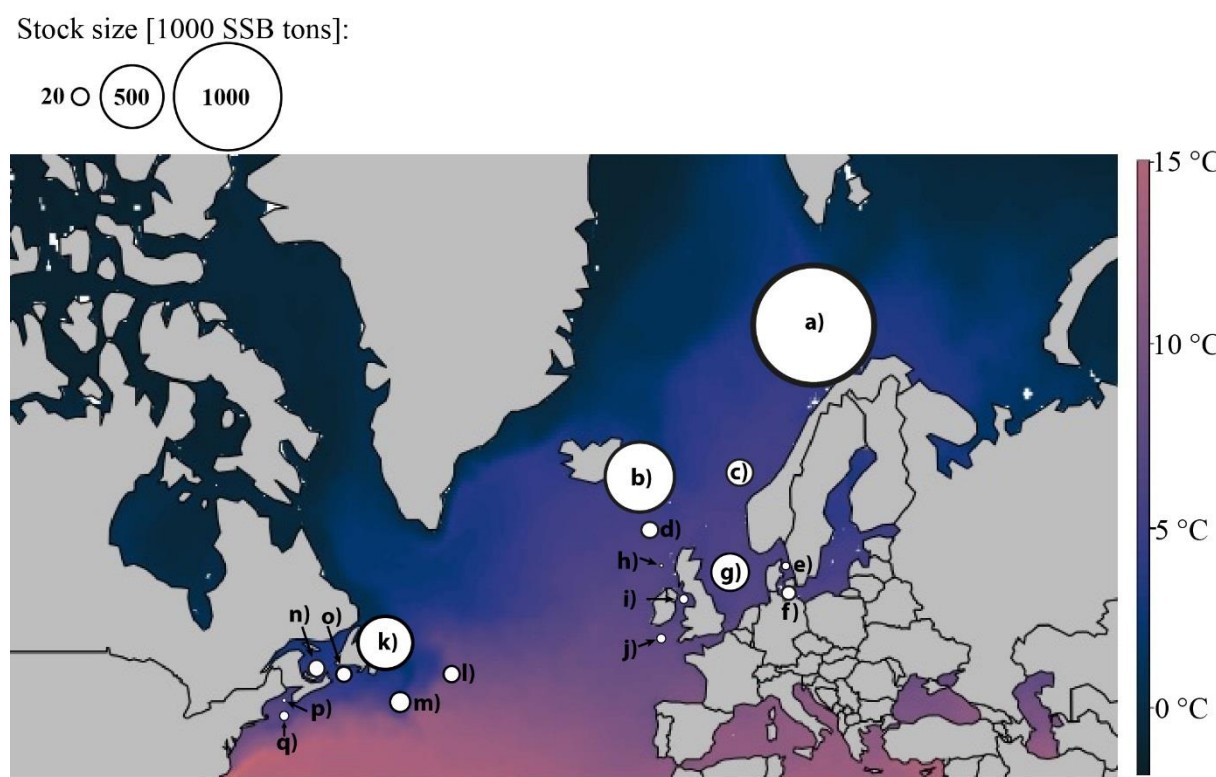


**Figure A1: Geographic map of average ambient sea surface temperature and stock size and locations according to their stock management area. Coloration of the map is according to ambient sea surface temperatures, which range for Atlantic cod between 3 °C in the Barents Sea (a) and 15 °C at the Flemish Cap (l). Spawning stock biomass (SSB) is highest from the North East Arctic (a) and Icelandic stock (b). The stocks investigated are: a) North East Arctic cod, b) Icelandic cod, c) (Norwegian) Coastal cod, d) Faroe**
**Plateau cod, e) Kattegat cod, f) Western Baltic cod, g) North Sea cod, h) West of Scotland cod, i) Irish Sea cod, j) Celtic Sea cod, k) Northern cod, l) Flemish Cap cod, m) Southern Grand Bank cod, n) Southern Gulf of St. Lawrence cod, o) Northern Gulf of St. Lawrence cod, p) Gulf of Maine cod, q) Georges Bank cod.**



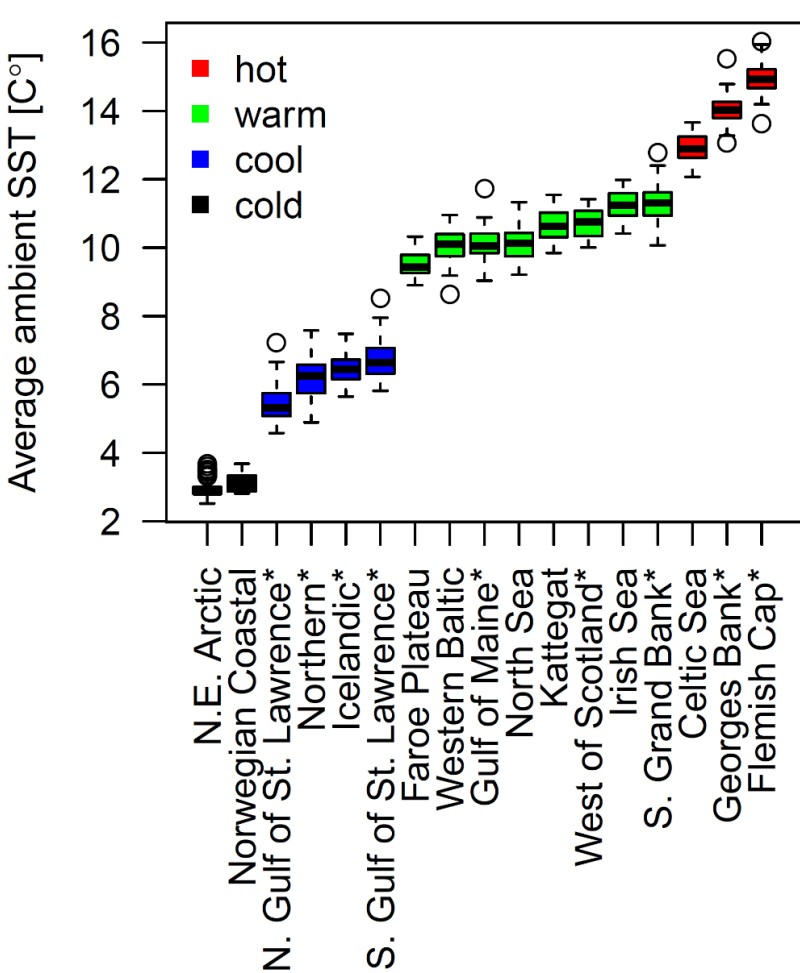

**Figure A2: Atlantic cod stocks grouping by their average ambient sea surface temperature (SST). Box plots show the temperature span of the time series available.**



**Appendix B: Stock assessment data and parameters**

For the 17 Atlantic cod stocks, we extracted time series on recruitment, SSB, age-specific abundance, fishing mortality, and life history traits (weight, probability of maturation and natural mortality) from publicly available assessment reports issued
by the different fisheries institutions responsible (ICES (International Council for the Exploration of the Sea, www.ices.dk), DFO (Department Fisheries and Oceans Canada, www.dfo-mpo.gc.ca), NAFO (Northwest Atlantic Fisheries Organization, www.nafo.int) and NOAA's Northeast Fisheries Science Center (www.nefsc.noaa.gov)). Stock specific number of age classes (5-13 classes), recruitment age (age 1-3) and the different length of time series available (17-68 years) was considered in the model (Table B1). Only the subset of data for which values of abundance, life-history traits and SST existed, were taken. P,
The probability of maturation (Eq. 4), was extended with the value of 1, if there was no data for older age classes available. F, the fishing mortality, was extended with the value for the last fish size class if needed. If the stock assessment reports did not contain all data needed, the time series of life history traits were completed by taking the average or using data from different reports (see comments, Table B1). Figure B1 shows the time trend relative to mean value of all data points (from all stocks) of total biomass (TB), spawning stock biomass (SSB), recruitment, recruitment per SSB (recruitment per capita),
spawner abundance, mean spawner age (probability of maturation and abundance weighted), meant maturation age (average age at which fish go to spawn for the first time in its life), mean spawner weight (probability of maturation and abundance weighted) and SST. All trends are significant with $p < 0.002$. All trends are negative, except for SST. Figure B1 shows the time trend in normalized SSB and fishing pressure of all stocks. Total fishing mortality was estimated as: $F(y) = -log \frac{\sum_1^A e^{-F(y,a)} N(y,a) W(y,a)}{\sum_1^A N(y,a) W(y,a)}$. Data were normalized to stock-specific average values.

Table B1: Data source, time-series length where all parameters were available, number of age classes, age of recruitment $a_R$, average ambient SST and data specifics for each of the 17 Atlantic cod stocks. Stars mark the western Atlantic cod stocks.

**Table B1: Data source, time-series length where all parameters were available, number of age classes, age of recruitment $a_R$, average ambient SST and data specifics for each of the 17 Atlantic cod stocks. Stars mark the western Atlantic cod stocks.**

| no | cod stock name | location | source | time-series length | age classes | $a_R$ | ambient $\overline{SST}$ [°C] | comments |
|---|---|---|---|---|---|---|---|---|
| 1 | NEA | I, II | ICES/AFWG | 1946 - 2014 (68) | 13 | 3 | 3 | |





| 2 | Norwegian Coastal | I, II | ICES/AFWG | 1984 - 2014 (30) | 9 | 2 | 3,1 | |
| 3 | *Northern Gulf of St. Lawrence | 3Pn, 4RS | DFO/CSAS | 1974 - 2014 (40) | 13 | 1 | 5,5 | |
| 4 | *Northern | 2J3KL | DFO/CSAS | 1983 - 2012 (29) | 11 | 2 | 6,2 | |
| 5 | *Icelandic | Va | ICES/NWWG | 1955 - 2015 (60) | 12 | 3 | 6,5 | weights data starts at age 3 |
| 6 | *Southern Gulf of St. Lawrence | 4T-4Vn | DFO/CSAS | 1971 - 2014 (43) | 11 | 2 | 6,7 | weights data starts at age 2 |
| 7 | Faroe Plateau | Vb1 | ICES/NWWG | 1959 - 2014 (55) | 9 | 2 | 9,5 | $F$ data starts at age 2 |
| 8 | Western Baltic | 22-24 | ICES/WGBAFS | 1994 - 2014 (20) | 7 | 1 | 10 | |
| 9 | *Gulf of Maine | 5y | NOAA/SAW | 1982 - 2014 (32) | 9 | 1 | 10,1 | $P$ - average from NEFSC |
| 10 | North Sea | IV: VIId, IIIa | ICES/WGNSSK | 1963 - 2016 (53) | 6 | 1 | 10,1 | |
| 11 | Kattegat | IIIa/21 | ICES/WGBAFS | 1997 - 2014 (17) | 6 | 1 | 10,6 | $F$ data for ages 1:4 (extended) |
| 12 | *West of Scotland | VIa | ICES/WGCSE | 1981 - 2014 (33) | 7 | 1 | 10,7 | |





| 13 | Irish Sea | VIIa | ICES/WGCSE/WGIrish | 1968 - 2013 (45) | 5 | 1 | 11,3 | *M* for age 0 accounted in age 1 |
|---|---|---|---|---|---|---|---|---|
| 14 | *Southern Grand Bank | 3NO | NAFO1 | 1959 - 2015 (56) | 10 | 3 | 11,3 | weights data starts at age 3 |
| 15 | Celtic Sea | VIIe–VIIk | ICES/WGCSE | 1971 - 2014 (43) | 7 | 1 | 13 | |
| 16 | *Georges Bank | 5z | NOAA/SAW | 1978 - 2014 (36) | 10 | 1 | 14,1 | *M* data taken from MRamp model version |
| 17 | *Flemish Cap | 3M | NAFO2,3 | 1972 - 2014 (42) | 8 | 1 | 15 | |

Abbreviations: ICES = International Council for the Exploration of the Sea, AFWG = Arctic Fisheries Working Group, DFO = Department Fisheries and Oceans Canada, CSAS = Canadian Science Advisory Secretariat, NWWG = North Western Working Group, WGBFAS = Baltic Fisheries Assessment Working Group, NOAA = National Oceanic and Atmospheric Administration, SAW = Northeast Regional Stock Assessment Workshop, WGNSSK = Working Group on Assessment of Demersal Stocks in the North Sea and Skagerrak, WGCSE = Working Group on Celtic Seas Ecoregion, WKIrish = Benchmark

Workshop on the Irish Sea Ecosystem, NEFSC = NOAA's Northeast Fisheries Science Center, NAFO = Northwest Atlantic Fisheries Organization; [1] (Rideout et al., 2017), [2] (González-Troncoso and Fernando González-Costas, 2014), [3] (A. Pérez-Rodríguez et al., 2016).





**Figure B1: Time trends of all data points of all 17 Atlantic cod stocks.**






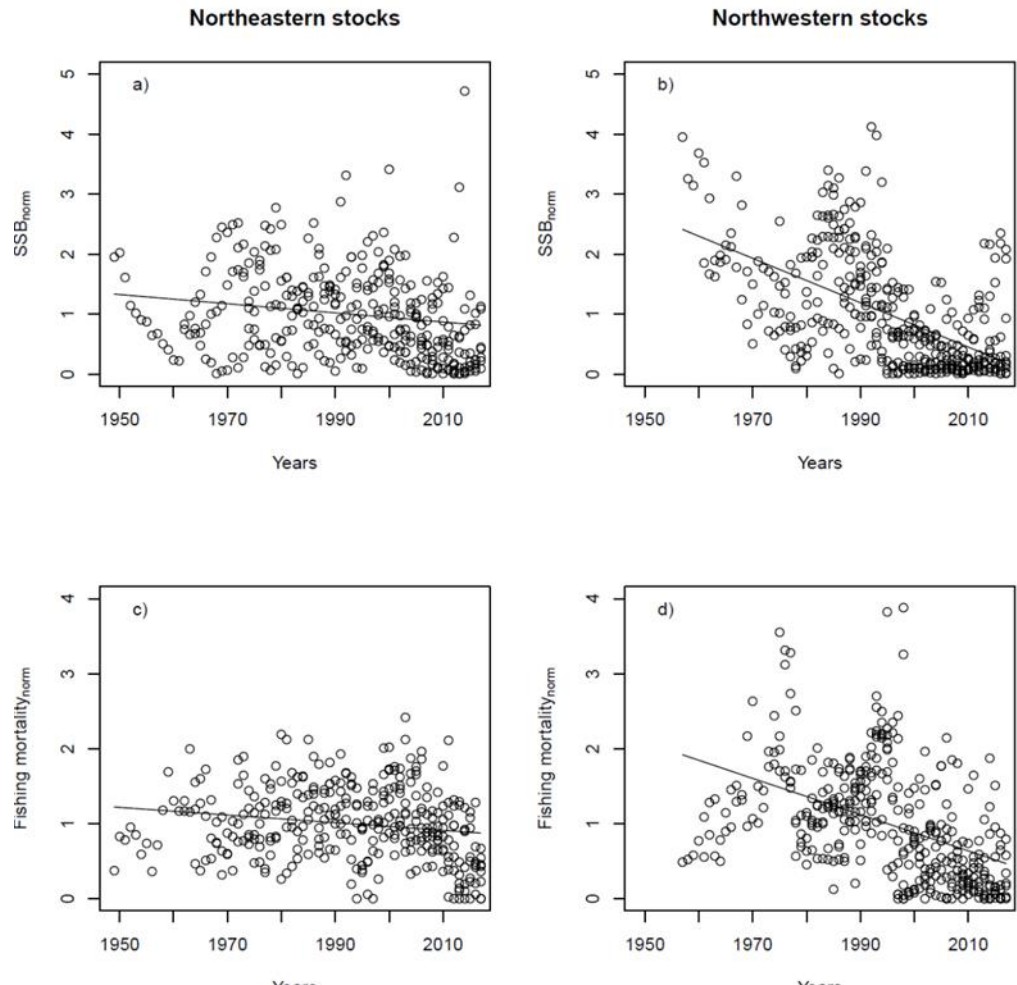

**Figure B2: Trends in normalized SSB and fishing mortality for Northeast Atlantic cod stocks (left plots) and Northwest Atlantic cod stocks (right plots).**

**Appendix C: The different stock-recruitment function components**

The relation between recruitment production and $SSB_N$ is captured by a sigmoidal function of spawner abundance $H(SSB_N)$, where recruitment per capita can, dependent on the parametrization, show both compensatory and depensatory dynamics of





different steepness, including predator-pit like Allee effects for recruitment per capita (C3). It therefore can capture the diversity in recruitment dynamics of the different Atlantic cod stocks (Eq. 5). $SSB_0$ is the position of the inflection point, where
the rate of recruitment production is highest, and $k$ is the function's steepness at $SSB_0$. See Fig. C1 for a schematic figure with all parameters explained.

The second factor, spwe, captures the effect of deviations in average spawner weight on recruitment production. Because $SSB_N$ is $SSB$ at average spawner weight, Eq. 6 depicts the year specific *deviations* from average spawner weight. It can be interpreted as a proxy for spawner fitness. While most conventional stock-recruitment model assume fecundity and egg production to
increase linearly with spawner weight (e.g. (Hilborn and Walters, 1992; Hutchings, 2005)), in this study we allow for a nonlinear dependence. We do this by introducing the exponent $c$, which quantifies the effect deviations from average spawner weight have on recruitment production. While $c = 1$ shows cases of linearity, as in most conventional stock-recruitment models using SSB as an aggregate, but also many models considering maternal effects (e.g. (Brunel, 2010; Shelton et al., 2015; Marteinsdottir and Thorarinsson, 1998)), $c > 1$ means that increasing spawner weight leads to an unproportional strong
increase in recruitment production (and vice versa decreasing spawner weight leads to unproportional strong decrease in recruitment; convex down) and $c < 1$ implies an unproportional slow increase in recruitment production with increasing spawner weight (convex up). Figure C1 shows the effect of parameter $c$ on the function (Eq. 6).

The third component of the stock-recruitment function is ambient SST, which is considered by a Lorentz function, $G$ (Eq. 7). Because recruitment response to SST anomalies can be stock-specific, the bell-shaped Lorentz function was chosen in contrast
to the conventional approach of modeling a monotonic temperature dependence effect (e.g. (Planque et al., 2003; Clark et al., 2003; Hilborn and Walters, 1992) though recent studies do recognize its time-variance (e.g. (Ottersen et al., 2013; Stige et al., 2013; Szuwalski et al., 2015; Olsen et al., 2011)). Laurel et al. (2017) recently applied a Lorentz function to describe growth dependence on SST and size in juvenile Arctic cod. A Gaussian function did not perform well, because some stocks show a nearly linear response to SST increase. Parameter $b$ of the Lorentz function describes the width of the temperature optimum
curve. See Fig. C1 for a description of each parameter. For five stocks, the Lorentz function finds a temperature optimum $T_0$, within the observed range of data (Fig. C2). For stocks where no temperature optimum in the range of observations was found, the whole temperature range in simulations localized on one side of the temperature curve (Table C1), e.g. in case of NEA and Faroe Plateau stocks, Fig. C2. Table C1 summarizes the temperature optima and width parameters of the fitted Lorentz function. The fitted temperature optima for recruitment production are within the documented range for Atlantic cod (e.g.
(Planque and Fredou, 1999; Drinkwater, 2005; Righton et al., 2010; Pörtner et al., 2001)), though depending on the physiological and/or ecological mechanism considered. Depending on the current position on the temperature optimum curve (red mark, Fig. C2), changes in future SST have a different impact on recruitment production. We estimated temperature





sensitivity (Table 1 in main text) as the ratio between recruitment at 0.5 °C SST increase and recruitment at present SST. In

Table 1 we show it as a percent change of projected recruitment from present SST (Eq. 7).

Considering the three effects on recruitment production, spnum, spwe and SST, the stock-recruitment function capturing all

different 17 Atlantic cod stocks, can thus be formulated as in Eq. 8.

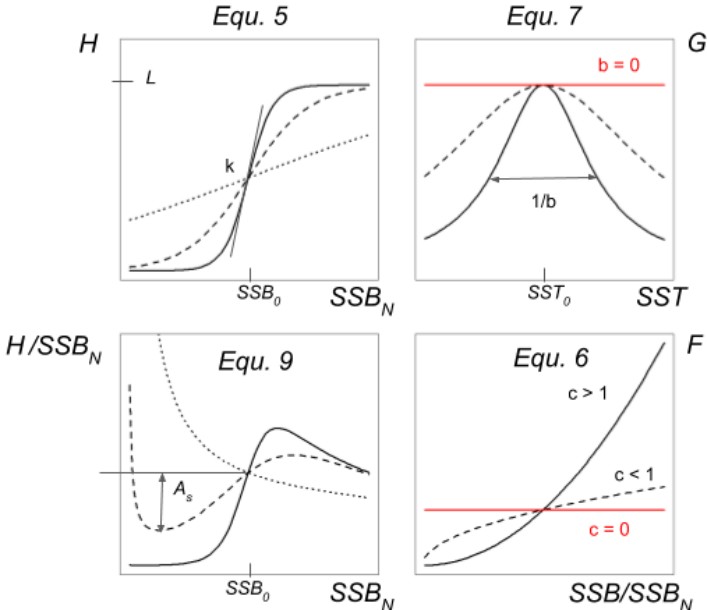

**Figure C1: Visualization of different equations and their parameters of the stock-recruitment function (Eq. 5-7) and the Allee effect**
**strength (Eq. 9).**

Figure C2 shows recruitment per capita as a function of average spawning temperature. Circles show the stock assessment

data, black dots indicate the simulated data with the respective best-fit stock-recruitment model. Note that number of spawners

at average weight (spnum) is always considered, in addition are changes in spawner weight (spwe) and/ or changes in SST.

For stocks with SST effect, the black line shows SST component of the stock-recruitment function. The red marks indicate the

most recent data.



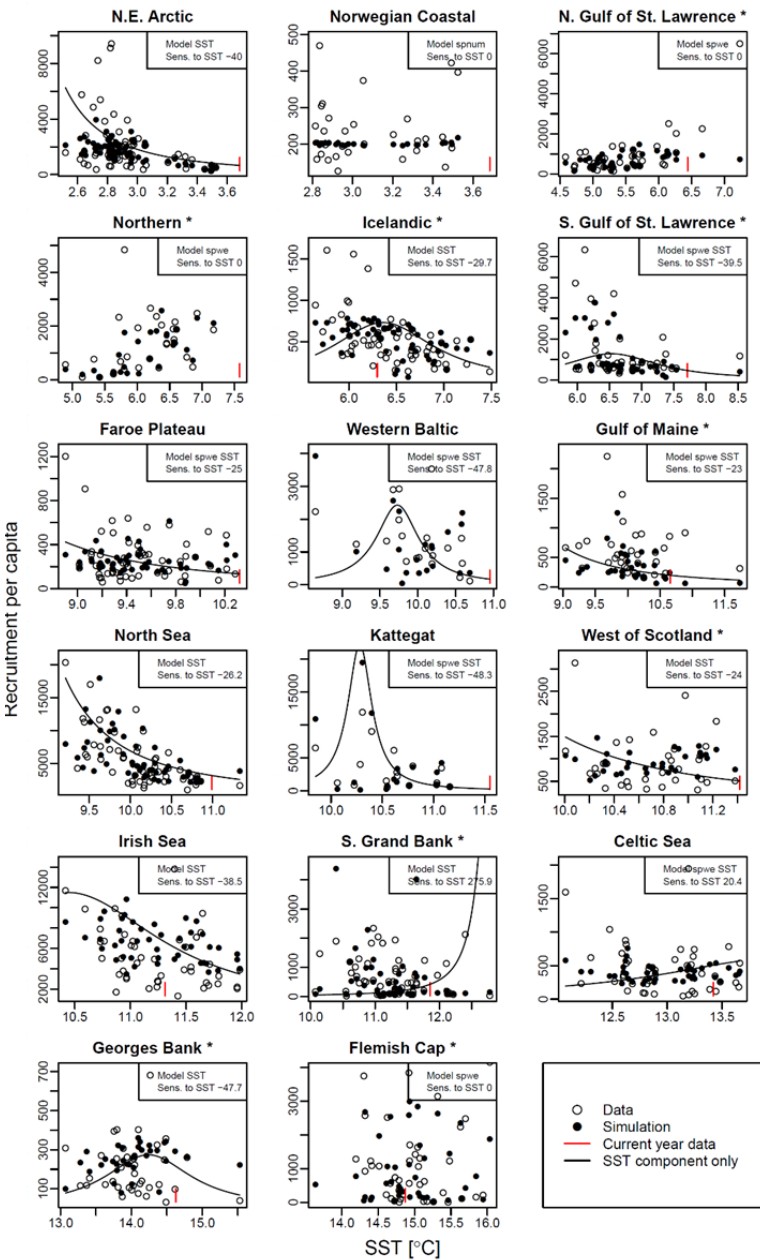

**Figure C2: Recruitment per capita ratios in relation to ambient SST. Dots show the stock-assessment data, black dots are simulated points. The black line indicates the stock-recruitment function with the SST function component only. The red mark indicates the most recent SST data point.**





**Table C1: The stock-specific model coefficients fitted for each Atlantic cod stock. SST$_0$ is the temperature optimum found.**

| No. | Atlantic Cod stock | Fitted stock-specific coefficients | | | | | |
|---|---|---|---|---|---|---|---|
| | | $L$ | $SSB_0/\overline{SSB}$ | $k$ | $c$ | $b$ | $SST_0$ |
| 1 | N.E. Arctic | 2012 | 0.7 | 1.07 | 0 | 11.52 | 1.97 |
| 2 | Norwegian Coastal | 197 | 1.42 | 1.05 | 0 | 0 | 2.82 |
| 3 | N. Gulf of St. Lawrence | 840 | 0.39 | 1.03 | 3.34 | 0 | 5.5 |
| 4 | Northern | 311 | 2.75 | 0.97 | 3.18 | 0 | 6.18 |
| 5 | Icelandic | 711 | 0.54 | 1.06 | 0 | 1.5 | 6.36 |
| 6 | S. Gulf of St. Lawrence | 1213 | 1.03 | 0.94 | 2.2 | 1.12 | 6.53 |
| 7 | Faroe Plateau | 235 | 0.42 | 1 | 0.71 | 7.82 | 7.09 |
| 8 | Western Baltic | 1334 | 2.72 | 1.02 | 6 | 2.95 | 9.73 |
| 9 | Gulf of Maine | 279 | 2.63 | 1.03 | 3.73 | 5.64 | 7.09 |
| 10 | North Sea | 6102 | 0.75 | 0.81 | 0 | 6.35 | 7.95 |
| 11 | Kattegat | 3000 | 0.78 | 0.94 | 3.28 | 7.05 | 10.28 |
| 12 | West of Scotland | 809 | 0.72 | 1.02 | 0 | 7.48 | 8.03 |
| 13 | Irish Sea | 7014 | 0.64 | 1.12 | 0 | 0.98 | 10.44 |
| 14 | S. Grand Bank | 175 | 2.15 | 1.03 | 0 | 5.35 | 12.83 |
| 15 | Celtic Sea | 377 | 3.49 | 0.76 | 4.37 | 0.74 | 14.04 |
| 16 | Georges Bank | 257 | 5 | 1.02 | 0 | 1.35 | 14.24 |
| 17 | Flemish Cap | 665 | 3.01 | 1.39 | 5.15 | 0 | 13.64 |

## Appendix D: Optimization procedure and model validation

The stock-specific coefficients L, c, b, T0, k, SSB0 (the Allee effect threshold, Table 1 in main text) were found by an optimization procedure, fitting time series of SSB. The system of equations (Eq. 1-8) was solved with the goal of obtaining a time series of SSB, here called SSBmodel, which maximizes the match with the SSB time series of stock assessment data, here called SSBdata. We chose SSBdata for parametrization, because of the smaller 95 % confidential interval of SSBdata (reported as SSBlow and SSBhigh in stock assessment reports) compared to those reported for recruitment data. The 95 % confidential interval was interpreted as data uncertainty and considered in Eq. D1 as: $E(y) = \ln\left(\frac{SSB}{SSB_{low}}\right)$. For stocks without a reported



confidence interval, the average uncertainty value among all stocks and years was used. The log-likelihood function used for optimization is given by:

$$q = -\frac{1}{2}\sum_{y=1}^{y_{max}} \frac{(\ln(\mathrm{SSB}_{model}(y)) - \ln(\mathrm{SSB}_{data}(y)))^2}{E(y)^2},$$ 
Eq. D1

with $y_{max}$ describing the maximum length of the time series (Table A1). Our goal was to find which combinations of the stock-recruitment components, spnum, spwe and/or SST minimize the divergence from the stock assessment SSB time series, while choosing spnum as the basis. Stock assessment SSB was treated as raw observed data, even though it is a result of a stock-

assessment model with its respective model assumptions.

As a first step of the optimization procedure the model was calculated using 5000 Latin hypercube samples. Latin hypercube sampling ranges for the parameters of the proposed stock-recruitment function were chosen wide enough to capture the observed range of stock-specific values ($L$: 0-2, the ratio of observed recruitment maximum and stock-specific average SSB ($R_{max}/\overline{\mathrm{SSB}}$), $\mathrm{SSB}_0/\overline{\mathrm{SSB}}$: 0-5, $k * \mathrm{SSB}_0$: 0-10, $c$: 0-6, $b$: 0-3 °C, $T_0$: 0-15 °C), so that maximum likelihood was not near the cube

surface. All hypercube samples were used to create initial probability density functions for the Monte-Carlo simulations.

The model parameter range was divided into a set of 200 bins, where within each bin a sampling probability $\rho_i$ was assigned based on maximum log-likelihood $q$ from subset of samples with a given parameter inside the bin: $\rho_i = \max(e^{-\frac{q}{q_0}})$, where $q_0$ is the normalization parameter – minimum log-likelihood of the currently best 200 samples. Such a dynamical adjustment of sampling probability allowed to avoid too sharp probability functions, while drastically increasing the performance compared

to a uniformly sampled Monte-Carlo simulation. The sampling probability was updated every 5000 samples. Model parameters were sampled independently according to the probability values for each bin.

The 10 most likely parameter combinations from the Monte-Carlo run were used as initial values for the Nelder-Mead algorithm to find a local minimum of the log-likelihood function. In the simulations of the log-likelihood function, Equ. D1, the initial cod population abundance in each stock was optimized instead of taking it from the first year of data time-series,

because of the high uncertainty usually associated with the first assessment year. Thus, initial conditions for the simulation were calculated using the age structure from the first assessment year multiplied by the adjustment factor. For each stock, the optimization procedure was carried out in four sets: spawner abundance only - fitting $L$, $k$, $\mathrm{SSB}_0$; abundance and spawner weight - fitting $L$, $c$, $k$, $\mathrm{SSB}_0$; abundance and SST - fitting $L$, $b$, $T_0$, $k$, $\mathrm{SSB}_0$; abundance and both spawner weight and SST - fitting $L$, $c$, $b$, $T_0$, $k$, $\mathrm{SSB}_0$. Model performance with the different stock recruitment effects was compared via model fitting

errors (RMSLE) and the Akaike information criterion, $AIC = \frac{2(m-q)}{n}$, where $n$ is time series length and $m$ is the number of parameters fitted. Dividing AIC by $n$ allows for comparison of model estimates for different time series lengths, as well as different parameter set sizes in the generic models within a stock. Table D1 shows the best model.





**Table D1: Comparison of the model fitting errors (RMSLE) and Akaike Information Criteria (AIC) for the different stock-recruitment components 1) spawner abundance, 2) spawner weight and 3) SST. The best, chosen model is shown in the right column.**

| No. | Atlantic Cod stock | RMSLE | | | | AKAIKE | | | | Best model |
|---|---|---|---|---|---|---|---|---|---|---|
| | | 1 | 1,2 | 1,3 | 1,2,3 | 1 | 1,2 | 1,3 | 1,2,3 | |
| 1 | NEA | 2.35 | 2.43 | 1.8 | 1.81 | 5.64 | 6.04 | 3.41 | 3.47 | 1,3 |
| 2 | Norwegian Coastal | 0.4 | 0.4 | 0.33 | 0.35 | 0.42 | 0.48 | 0.49 | 0.57 | 1 |
| 3 | Northern Gulf of St. Lawrence | 1.68 | 0.86 | 1.11 | 0.86 | 3.01 | 0.99 | 1.53 | 1.09 | 1,2 |
| 4 | Northern | 0.85 | 0.41 | 0.65 | 0.35 | 0.99 | 0.5 | 0.82 | 0.59 | 1,2 |
| 5 | Icelandic | 1.12 | 1 | 0.83 | 0.83 | 1.38 | 1.16 | 0.89 | 0.92 | 1,3 |
| 6 | Southern Gulf of St. Lawrence | 3.58 | 0.68 | 1.81 | 0.59 | 12.97 | 0.69 | 3.54 | 0.67 | 1,2,3 |
| 7 | Faroe Plateau | 1.43 | 1.22 | 1.1 | 1.05 | 2.2 | 1.67 | 1.43 | 1.35 | 1,2,3 |
| 8 | Western Baltic | 0.79 | 0.78 | 0.58 | 0.45 | 1.01 | 1.09 | 0.91 | 0.87 | 1,2,3 |
| 9 | Gulf of Maine | 1.18 | 1.08 | 1.09 | 0.99 | 1.63 | 1.47 | 1.54 | 1.41 | 1,2,3 |
| 10 | North Sea | 1.84 | 1.43 | 1.09 | 1.1 | 3.55 | 2.24 | 1.42 | 1.46 | 1,3 |
| 11 | Kattegat | 1.5 | 1.5 | 1.11 | 0.58 | 2.71 | 2.81 | 1.89 | 1.11 | 1,2,3G |
| 12 | West of Scotland | 1.64 | 1.61 | 1.34 | 1.34 | 2.94 | 2.87 | 2.16 | 2.22 | 1,3 |
| 13 | Irish Sea | 3.05 | 3.05 | 2.79 | 2.79 | 9.5 | 9.54 | 8.04 | 8.11 | 1,3 |
| 14 | Southern Grand Bank | 2.14 | 2.14 | 1.69 | 1.7 | 4.73 | 4.75 | 3.05 | 3.13 | 1,3 |
| 15 | Celtic Sea | 1.47 | 1.41 | 1.4 | 1.27 | 2.33 | 2.23 | 2.23 | 1.93 | 1,2,3 |
| 16 | Georges Bank | 0.77 | 0.77 | 0.64 | 0.64 | 0.81 | 0.86 | 0.74 | 0.79 | 1,3 |
| 17 | Flemish Cap | 3.62 | 2.56 | 3.64 | 2.69 | 13.27 | 6.77 | 13.5 | 7.55 | 1,2 |
| | Stocks mean | 1.73 | 1.37 | 1.35 | 1.14 | 4.06 | 2.72 | 2.8 | 2.19 | 1,2,3 |


Figures D2-4 show the comparison between the simulated estimates based on the respective best-fit stock-recruitment function and the stock assessment data.



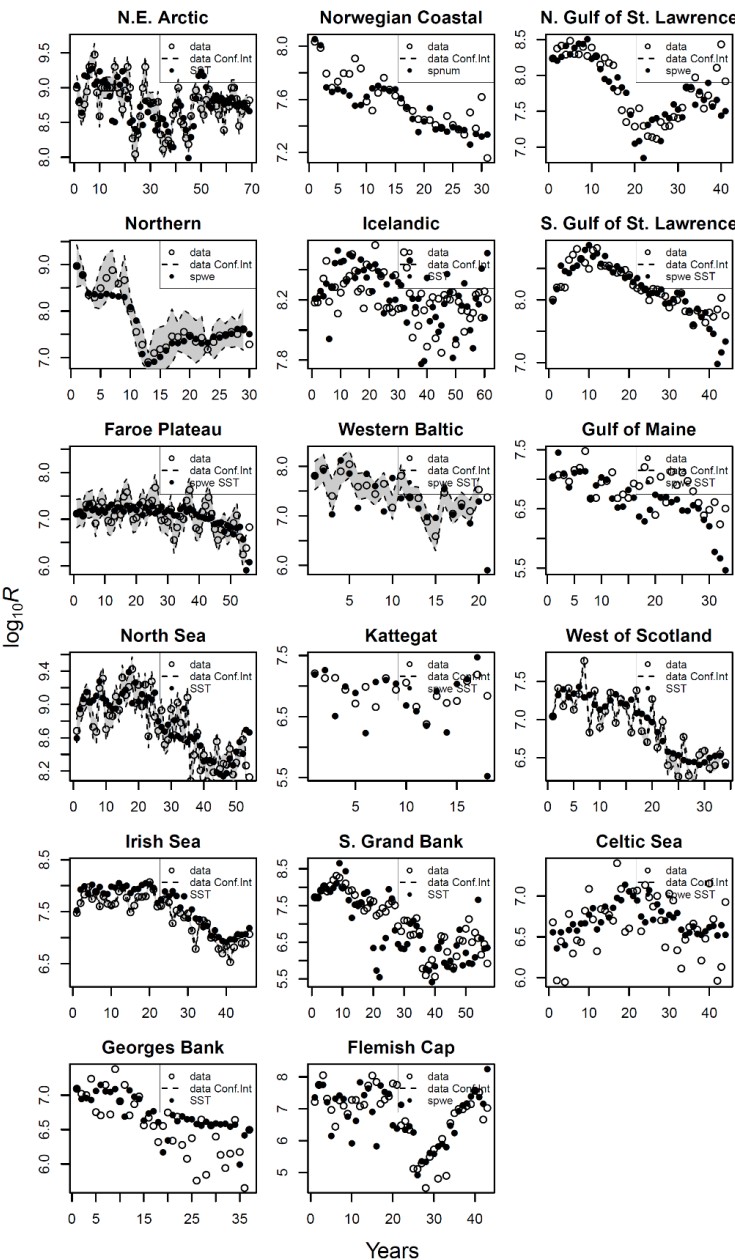

**Figure D1: Comparison of the simulated recruitment data points (black dots) with the best fit stock-recruitment model and the stock-assessment data (circles). Where available, the 95 % confidence interval of the stock assessment data is shown in grey.**




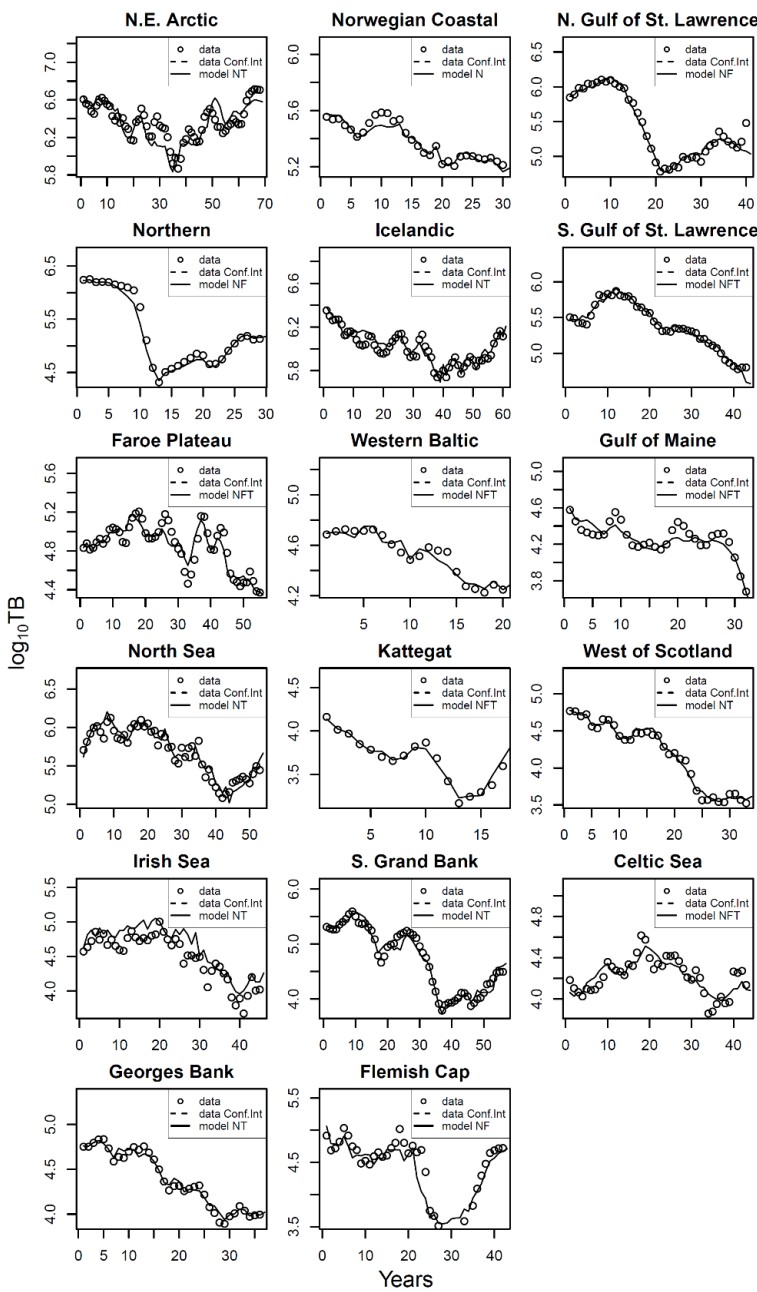

**Figure D2: Comparison of the simulated total biomass (TB) data points (black dots) with the best fit stock-recruitment model and the stock-assessment data (circles).**





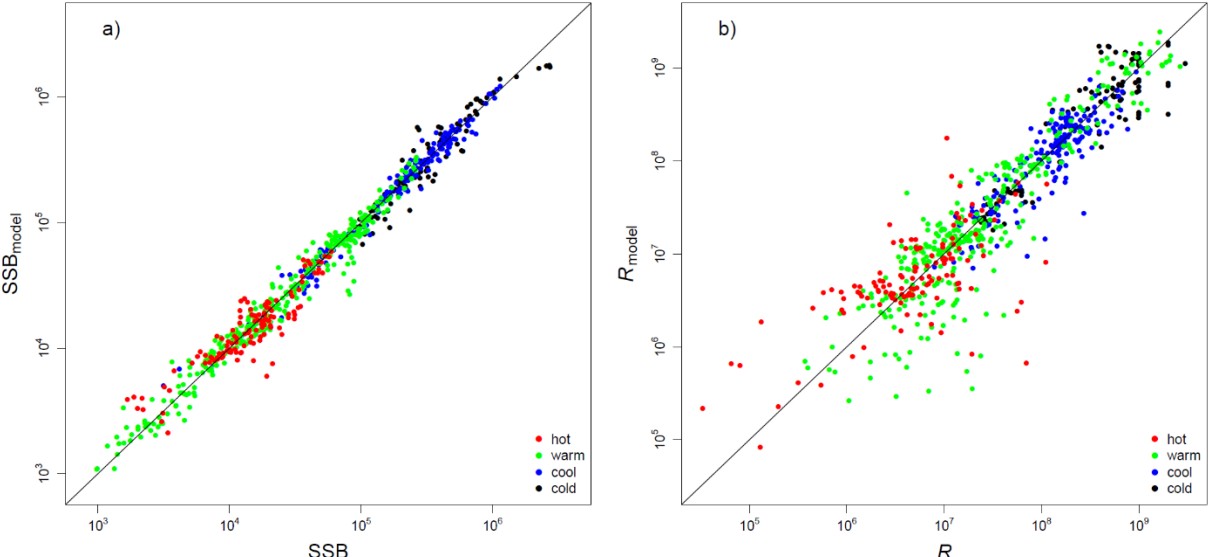

**Figure D3: Comparison between simulated estimates (y-axis) for SSB (left) and R (right) and the stock-assessment data (x-axis). Colors indicate the grouping according to ambient water temperature (Fig. A2).**

**Appendix E: Changes in population characteristics**

Appendix E contains different stacked plots from stock-assessment data. Figure E1 shows how body weight deviated from

average over time within each Atlantic cod stock and age class.



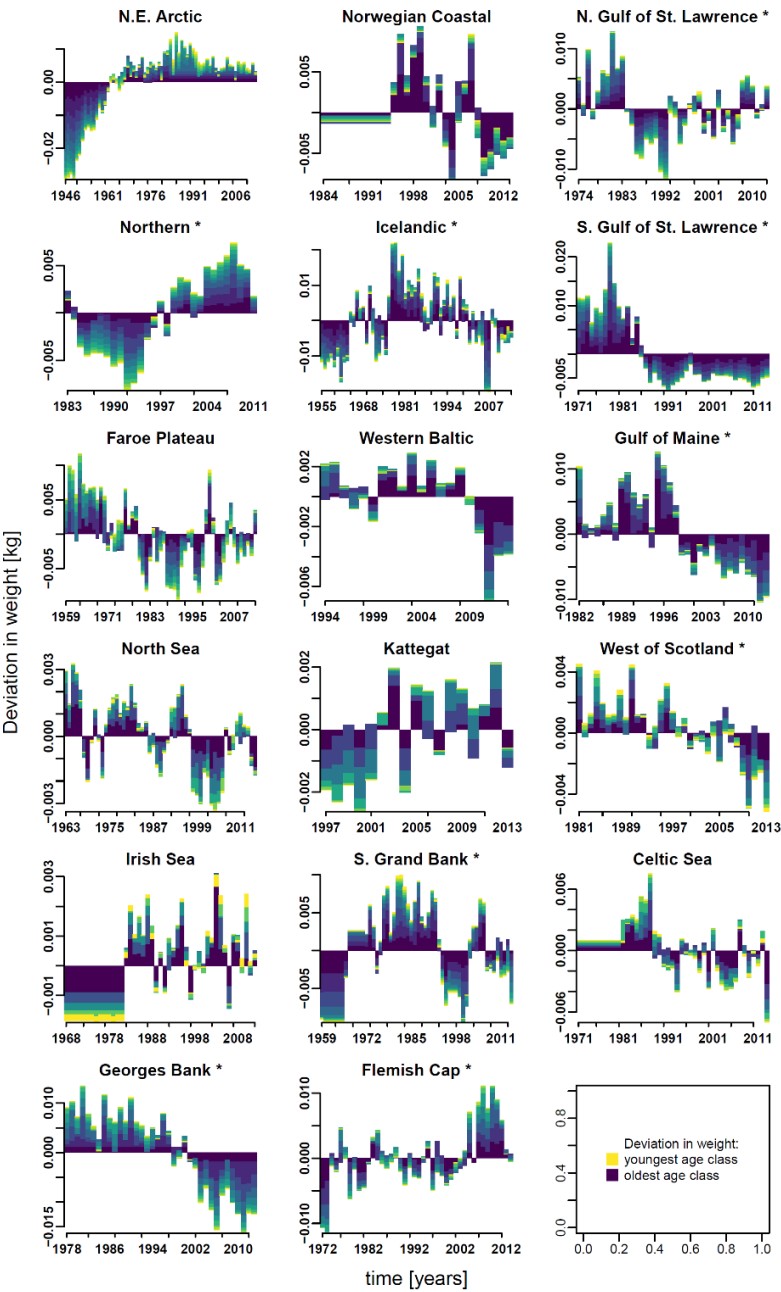

**Figure E1: Change in body weight relative to mean weight, for each age class according to stock-assessment data. Different colors indicate the different age classes (yellow: youngest age class, dark blue: oldest age class).**



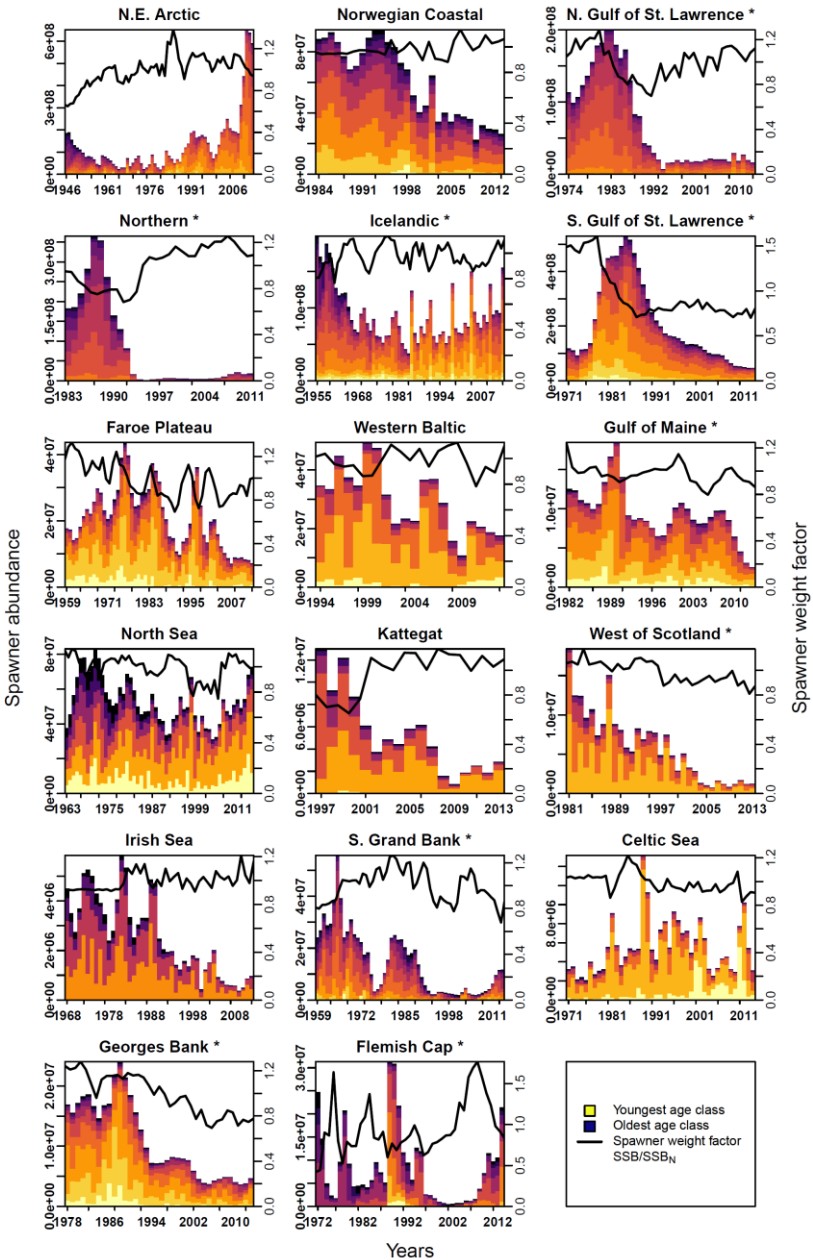

**Figure E2: Change in spawner abundance (number of mature fishes) for each age class according to stock-assessment data. Different colors indicate the different age classes (yellow: youngest age class, dark blue: oldest age class). The solid line shows weight of the average spawner, which reflects the weight of the most abundant spawning class in a specific year (right y-axis).**



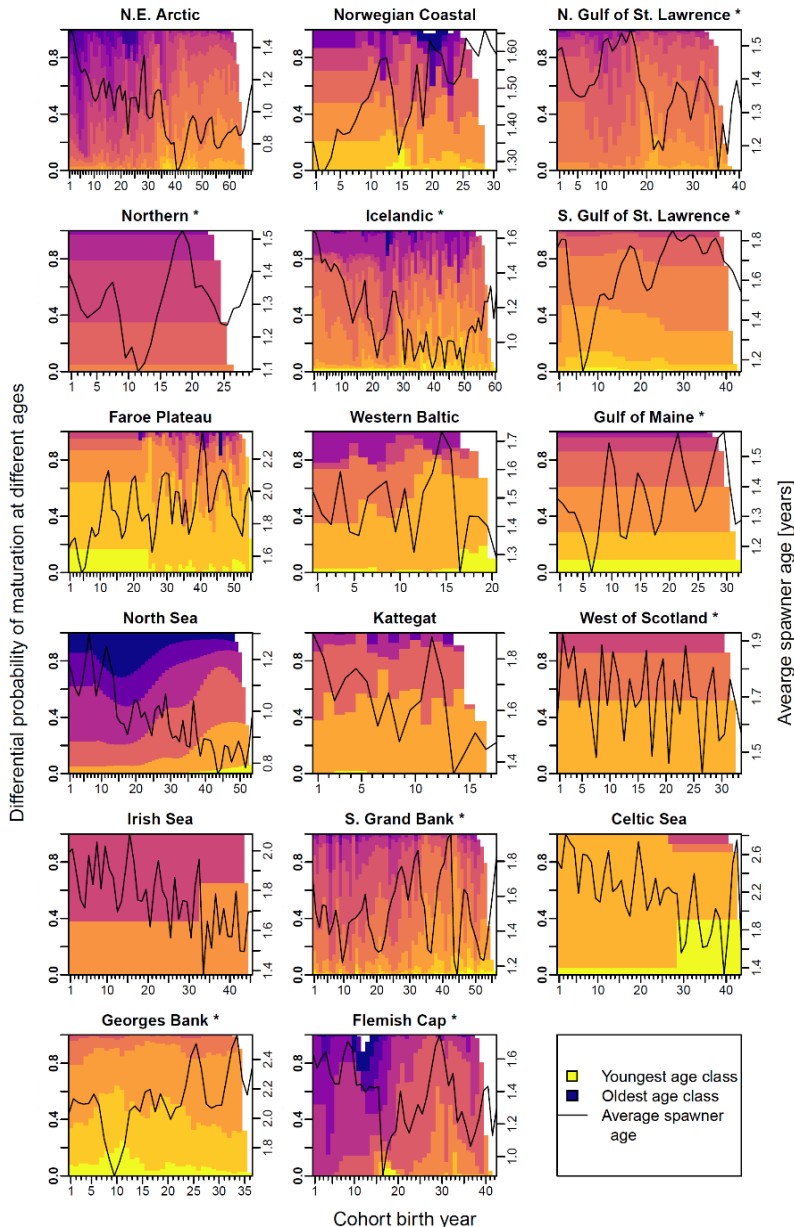

**Figure E3: Change in the probability of maturation for each age class according to stock-assessment data. Different colors indicate the different age classes (yellow: youngest age class, dark blue: oldest age class). The black line shows the average spawner age (right y-axis).**



**Author contribution**

A-M.W., N.V. and A.V. jointly conceptualized the research and applied for research funding. N.V. and A.V. took lead in
developing the methodology, programming and statistical analysis. A-M.W. collected the data, provided the literature and
wrote the first draft. A-M.W., N.V. and A.V. jointly created visualizations of the results and finalized the manuscript.

**Competing interests**

The authors declare that they have no conflict of interest

**Acknowledgements**

This research is a deliverable of Nordic-Russian collaboration project "A multidisciplinary approach to anticipate critical
regime shifts in ecosystems" (TerMARisk), funded by Nordforsk [grant number 81513]. A.-M.W. was further funded by the
project OILCOM from the Research Council of Norway [grant number 255487]. The authors wish to thank T. Vasiliev for
providing the computational infrastructure for the Monte-Carlo calculations and J. A. Hutchings, M. Eikeset, Ø. Langangen
and A. Richter for helpful discussions. This paper is dedicated to the memory of Jeffrey A. Hutchings.

**Data availability statement**

All data used here is publicly available without special permission. No new data was generated. Biological data was retrieved
from stock assessment reports from the different responsible fisheries institutions (For Atlantic cod stocks in waters of the EU:
ICES (International Council for the Exploration of the Sea, www.ices.dk), for cod stocks in Canadian waters: DFO
(Department Fisheries and Oceans Canada, www.dfo-mpo.gc.ca), for cod stocks in waters of the U.S.: NAFO (Northwest
Atlantic Fisheries Organization, www.nafo.int) and NOAA's Northeast Fisheries Science Center (www.nefsc.noaa.gov).
Temperature data was Time series of sea surface temperature, SST, from each stock's geographical location was extracted
from NOAA's Earth System Research Laboratory, Physical Sciences Division (NOAA_ERSST_V4 data,
www.esrl.noaa.gov/psd/), according to each Atlantic cod stock's location. Appendix B gives a full description of the data and
sources used.
The programmed R code is provided by the authors.



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
