# Peer review of "Spawner weight and ocean temperature drive Allee effect dynamics in Atlantic cod, *Gadus Morhua*: inherent and emergent density regulation"

_EGUsphere, 2022_

## Author Response (AR1)

The p.<page> l.<line> comments refer to a revised manuscript version (in a track changes mode).

**Authors Response to Editor comment 2**

**EC2:** The revisions to the manuscript have adequately addressed the comments of the reviewers. This manuscript is an opportunity to link fish with environmental variables and biogeochemistry. To ensure the relevancy of the analyses to the readership of e readership of "Biogeosciences", I am requesting you add one more paragraph to the section "4.2 Linking temperature to biogeochemical processes". In this additional paragraph, please describe exactly what environmental and biogeochemical variables you would ideally have to do a better analysis and to possibly reveal the underlying mechanisms. The section 4.2 is good - one (or two if you prefer) more paragraphs would complete the linkage to biogeochemistry. In an ideal world, what variables, on what spatial (horizontal, vertical) and temporal resolution (e.g., daily, monthly), and for types of years (e.g., low and high population abundance) would allow you to advance your correlation-oriented analysis? In addition to salinity, chlorophyll, etc., perhaps you could use some rates variables related to productivity to indicate food or can use transport information? Please think big and try to be specific. You can start the paragraph with a caveat that you are speculating. The next version of the manuscript will be evaluated by me.

**Authors:** We elaborated more on this point and added two additional paragraphs into the discussion **"4.2 Linking temperature to biogeochemical processes"** on p.28, l.544-562:

"…conditions are important. However, before linking our study to other (biogeochemical) models, we would rather suggest more experimental work to study the effects of environmental variables in relation to population density and abundance. This would fundamentally advance the research on Allee effects, moving from theory to evidence, about the existence of Allee effects and their mechanisms.

In particular, we would suggest more laboratory analysis in closed mesocosms to analyze all biological factors that influence net recruitment (i.e. hatching rate, survival, egg number, feeding rate) in relation to temperature and abundance. These data would allow to introduce more mechanisms in the model, including those which use the same factors. For example, SST can affect both probability of eggs hatching in water and abundance of food. Food can serve as a separate mechanism in the recruitment function, in case of more detailed data on food availability for different age classes (i.e., plankton, sprat and etc.). Weight deviations from age-specific average would remain a life trait separate from food availability in a certain year.

To see when and where lab conditions are met in the real world fish tagging with data loggers could be of big importance in addition to existing regular grid monitoring of environmental

conditions in the sea. Ideally, obtained fish life-history profiles of environmental variables could then be used to construct rates, specific averages and other features for correlation analysis with individual fish properties such as weight, fecundity and other. More data on individual fish movements and environmental conditions are required for this approach (e.g., ambient water temperature, pressure, etc. for fishes of different age classes either calculated by detailed tracking of fish movement or measured with data loggers attached to fishes of each age class each year). Individual fish behavior during lifetime e.g., spawning events can be also obtained with fish track data and correlated with obtained life-history profiles of its ambient environmental factors. The latter would be ideal for analyzing Allee effects in general and being able to distinguish between responses to ambient environment conditions and to population abundance."

**Authors Response to Reviewer 1**

**R1:** This study investigated the effects of spawner abundance, weight, and temperature on recruitment dynamics for 17 Atlantic cod stocks. Specifically, the authors examined the Allee effect, which is when recruitment per capita declines at low population density. The authors found most stocks demonstrated an Allee effect related to spawner abundance; however, Allee effect strength varied among stocks when considering water temperature and spawner weight.

Overall, the manuscript is on an important topic considering the status of many global marine fish stocks and increasing anthropogenic pressures. The methods and equations are clear, the statistical analyses appear sound, and the level of detail is appreciated. There are several technical comments that should be addressed prior to publication

**Authors:** We thank reviewer R1 for these encouraging comments. In the revised manuscript we followed the technical suggestions to improve it.

**R1-1:** Acronyms for "DFO" and "IPCC" are not capitalized in citations throughout.

**Authors:** thanks, DFO and IPCC are capitalized on p.2 l.39, p.26 l.470, p.30 l.581.

**R1-2:** Figures need to be higher quality with larger text, some are blurry or too small to read.

**Authors:**

Unfortunately, the figures lost quality because of introducing them into word docx file as images. All figures are now reinserted into docx file in svg/emf vector format to preserve the best quality of original figures, and texts are readable everywhere. Each figure will be provided

for publication also as a vector image in a pdf format to provide the best quality. Figures 2, A2 and D3 color schemes were changed to meet colorblind requirements.

**R1-3:** Line 120: Our objective is [to describe the data]?

**Authors:** thank you, change made on p.6 l.122 "the description of the data" to "to describe the data".

**R1-4:** "Table A1" does not exist in any table title, only in the text.

**Authors:** yes, this is because the table was moved to appendix B, we have corrected correspondingly in the text "Appendix A: Table A1" replaced by "Appendix B: Table B1" on p.7 l.154.

**R1-5:** Line 153: no need to repeat "sea surface temperature" since you introduced the acronym above.

**Authors:** thanks, extra text removed through the text replacing "sea surface temperature, SST" with "SST".

**R1-6:** Line 156: describe what a(subscript R) is.

**Authors:** R - corresponds to recruitment, $a_R$ - age of recruitment is being introduced above on p.7 l.153-154.

**R1-7:** The numbered equations are good, but please ensure you are referring to each equation in the text as well.

**Authors:** indeed, all of the equations were referred to only in the Appendix D. Now we also added a sentence in the main text part of the optimization procedure "The model of cod population dynamics, formulated as a system of Eqs.1-8, has.." on p.11, l.269 and some of them are also referred to elsewhere.

**R1-8:** Line 268: should be "affected" not "effected"

**Authors:** thanks, "effected" changed to "affected" on p.14. l.312.

**R1-9:** Lines 175-177 introduces an important intention/potential use of this research (not intended to replace existing stock-assessment practice etc). I would encourage a clear statement like this in the introduction as well when explaining the significance/importance of this study.

**Authors:** thank you, we added the following paragraph to the introduction on p.6 l.130-131 "While our approach to model development is not intended to replace existing stock-assessment practice, it can be used to reveal potential mechanisms affecting population dynamics."

**R1-10:** Line 392: the statement on how this is the first study showing Allee effect for every individual stock should be stated in the introduction as well.

**Authors:** we added to the introduction the following sentence on p.6 l.132-133 "In this study each individual stock is for the first time analyzed for the potential of Allee effect dynamics with the same approach and criteria."

**R1-11:** Table D1: Acronym for Akaike Information Criteria is not consistent in the text, title, and table.

**Authors:** thank you, we have made it consistent to AIC in the table D1 and in the text on p.11 l.275.

**Authors Response to Reviewer 2**

**R2**: The paper by Winter et al. develops a novel stock-recruitment function that captures the diversity of density dependencies (compensation, Allee effect) of recruitment production and disentangles the effects of spawner abundance, spawner weight and temperature on recruitment dynamics. The developed model is fitted to empirical time series to identify Allee effects and their potential drivers for Atlantic cod. Overall, the paper contributes to the emerging recognition of dynamic Allee effects that can interact with the environment, and as such, it addresses relevant scientific questions.

Nonetheless, some parts of the paper require careful rewriting or reorganization to make it easier to follow. See my specific comments below.

**Authors:** Thanks reviewer R2 for the detailed suggestions. In the revised manuscript we worked on rewriting the text, considering all these specific comments, and added the suggested model overall structure scheme.

**R2-1**: 46 Check spacing after '(SSB)' and before 'the biomass'.

**Authors:** corrected.

**R2-2**: 52, 101... (many occasions) Check the use of parentheses in citations throughout.

**Authors:** thank you, extra parentheses removed throughout the text.

**R2-3**: Figure 1 & 113-118 It is not usual to include results of an exploratory analysis in the Introduction. Could you motivate the study setup in another way and have these in Materials and Methods or Appendix?

**Authors:** thanks, we thought of stock assessment data as literature data to be referenced to in the introduction. However, since we base on our own data analysis, it is indeed more logic to place it as a part of our work. We have inserted the following paragraph and the figure 1 in the beginning of the Results section on p.12, l.281 as follows: "Stock assessment data analysis resulted in a density-plot of "raw" stock assessment data from all 17 cod stocks (Fig. 1) which suggested the highest probability for a decrease in recruitment per capita at below average spawner abundance, indicating patterns of an Allee effect (Fig. 1 a). Interestingly, most stocks have been prevalent in the area of the Allee effect threshold (Fig. 1 a, white area), where SSB degradation can have strong repercussions for population and management. In particular at low spawner abundance, below the Allee effect threshold, recruitment per capita ratios are accompanied by temperatures above the average experienced SST (Fig. 1 b, red shading), as well as a below average spawner weight (Fig. 1 c, blue shading). This led us to a hypothesis that spawner weight and SST could have effects on recruitment production at low abundance, which is further tested by the developed model."

While instead in the introduction on p.6 l.120 we say, "Based on literature and preliminary stocks data analysis we hypothesize that consideration of spawner abundance, spawner weight and SST as components of the stock-recruitment function, should give insight on Allee effect dynamics in Atlantic cod.".

**R2-4**: 101 Also, Fig. A1 could be cited in M&M, instead of citing it in the Introduction.

**Authors:** on p.4 l.101-102 of the introduction we changed the phrase "The different stocks are located in the North Atlantic Ocean (see Appendix A: Fig. A1) ), where direction and intensity.." to "In the North Atlantic Ocean direction and intensity.. " and moved the citation of Fig.A1 to M&M to p.7, l.136 as follows "For the 17 Atlantic cod stocks, that are located in the North Atlantic Ocean (see Appendix A: Fig. A1), we extracted time series".

**R2-5**: 147-148 I assume the age class is a discrete variable. I would rather say its value belongs to the set {1,...,A}.

**Authors:** thanks, corrected on p.7 l.152.

**R2-6**: 155 What does 'R' stand for in eqn. (1)? Is it recruitment (line 257)?

**Authors:** yes, thanks, on p.7 l.159 the change is made from "recruitment production to be.." to "recruitment production function R to be.."

**R2-7**: 175-177 Should this rather be included in the Discussion?

**Authors:**

we have moved this phrase to the end of the discussion on p.31 l .617-619.

**R2-8**: 168-169 & 178-184 I got confused about the description of the basic demographic component. Line 169 states that 'spnum' stands for the abundance of spawners. But, in lines 182-183 you elaborate whether spnum captures mainly the impact of spawner abundance. Is the paragraph in lines 178-184 supposed to describe the 'basic demographic component'? Is this basic demographic component given by eqn. (4)? Or is it given by eqn. (5)? Please, make clear what 'spnum', SSB_N and the demographic component are.

**Authors:**

Sorry for the confusing description. We used words "spnum", "spwe" and "SST" as short labels for the models (especially needed for the figures) that take into account effects of spawner number, spawner weight, SST, correspondingly.

SSB_N - is a value calculated similar to SSB, using age-specific historical average spawner weights instead of actual weights and is defined by the Eq.4. It is used instead of SSB in the stock-recruitment function to isolate spawner abundance effect from spawner weight effect (see the new general model scheme Fig.C2).

Demographic component of the stock-recruitment function is the function - $H(SSB_N)$ defined by the Eq. 5.

In the revised version we use these short abbreviations only for labeling the models to designate combinations of analyzed effects. We have revised the section 2.3 "Stock-recruitment function with separable effects" on pp.8-10 to make its description clear.

**R2-9**: 196 the effect 'of'?

**Authors:** corrected.

**R2-10**: 203 Introduce eqn. (6) earlier and not separate from the text. For example, 'The second factor, spwe, captures the effect of deviations in average spawner weight on recruitment production and is defined by $F(y) = (\text{SSB}(y)/\text{SSB}N(y))$   (6).'

**Authors:** thank you, the equation introduced as suggested earlier in p.9. l.212.

**R2-11**: 204 Similarly, introduce the equation for the Lorentz function immediately after first mention.

**Authors:** corrected, Lorentz function introduced on p.10. l.227 after its first mention.

**R2-12**: 211-217 (at least) If these are findings after model fitting and not data preprocessing, should they be reported in the Results section instead?

**Authors:** yes, indeed. We moved this paragraph to p.14 l.318-326 of the Results.

**R2-13**: 222 Please include in the text and introduce earlier. What was SST_0? You're talking about T_0 in the text above eqn. (7).

**Authors:** this was a typo, on p.10, l.228, p.11.253, 270 "$T_0$" changed to "$\text{SST}_0$".

**R2-14**: 211 Given that there seem to be only two parameters for G, why not introduce both of them in the main text.

**Authors:** We have added right after the equation 7 on p.10 l.228 both parameters with the following "where $\text{SST}_0$ is SST optimum value, and b – sensitivity to SST.".

**R2-15**: 226-227 Often, one prefers not to start a sentence by a symbol. Consider throughout the paper.

**Authors:** thank you, corrected throughout the text.

**R2-16**: 240-242 Please modify in a similar way as suggested above.

**Authors:** done.

**R2-17**: 244 Now there's also SSB_0 to optimize. Should it also be listed earlier (line 228)?

**Authors:** Thank you, we forgot to list this parameter. This is corrected now on p.11, l.254.

**R2-18**: Sections 2.2-2.3 A schematic figure of the overall structure of your model would be nice.

**Authors:** We have added the scheme of overall model structure as Appendix C: Fig. C2. instead of the extra duplicating text about the model. Former Fig.C2 became Fig.C3.

[Figure]

**R2-19**: 258 I didn't find figure D4 in the Appendix. The last 25 years here mean..?

**Authors:** thanks, the deletion of one figure in D caused the shift in cited numbers, on p.47 l.766 "Fig. D2-4" is changed to "Fig. D1-3".

We changed the phrase "Within the last 25 years, all stocks experienced strong declines in SSB" to "All of the stocks cod experienced strong declines in SSB with their historical minima observed after 1990 except for NEA cod stock.".

**R2-20**: Figure 2 The figure is great, but the font size could be bigger and, especially, the resolution of the graphics should be higher.

**Authors:** Unfortunately the figures lost quality because of introducing them into word docx file. All figures are now reinserted into docx file in svg/emf vector format to preserve the best quality of original figures. Texts are now readable in all Figures. Each figure will be provided for publication as vector images in a pdf format to provide the best quality.

**R2-21**: 265 But for many of the stocks in Fig A2, the average SST seems to be over 11? For example, Flemish Cap, Georges Bank.

**Authors:** Very true, corrected to "Average ambient sea surface temperature ranges between 3 °C and 15 °C (Appendix A: Fig. A2)…" on p.14, l.309.

**R2-22**: 266 There is no Figure A1 in Appendix B. However, Fig. B1 illustrates trends and p values for individuals stocks over the whole time series, should you refer to it later (in line 267)?

**Authors:** yes, the reference to figure B1 is corrected on p.14 l.311.

**R2-23**: 276-277 I'm not sure if I understood your reasoning. Please, expand/say in another words.

**Authors:** we improved this paragraph on p.15 l 329-334 as follows: "For example, when fishing pressure of the NEA cod started to decline as part of a management plan, the ocean water was also cooler (triangles symbols, Fig. 2) and thus the stock growth temperature component had the highest value (Appendix C: Fig. C2). This could be the reason behind the increase and recovery of its SSB. Further decline in fishing mortality happened concurrently with ocean warming and likely overweighted the negative response to SST, and the stock continued to grow."

**R2-24**: 291 Check consistency with the description in Section 2.3.

**Authors:** we have made it consistent through out the text designations of the 3 components of SR-function - H(), G(), L() functions and corresponding model designations by significant effects (spnum, spwe, SST).

**R2-25**: 308-310 Describe the symbols of your figure in the legend instead. Refer here only to your model (components)?

**Authors:** we have changed this to "We consider that spnum component of SR-function should reflect the inherent density-dependence regulation".

**R2-26**: Overall, the text should be readable without the figures and the figures should be described in their legends. For example, instead of 'A high goodness of fit for total biomass (TB) and recruitment (R) time series is shown in Appendix D: Fig. D2-4).' (lines 257-258), could you say 'We obtained a high goodness of fit for total biomass (TB) and recruitment (R) time series (Figs D2-4).'?

**Authors:** thanks for the suggestion, we corrected accordingly on p.14, l.299-300 and header of table D1, captions of Fig.2, 3, C2, D1-3. Figures 2, A2 and D3 color schemes were changed to meet colorblind requirements.

**R2-27**: Since the Allee effect threshold SSB_0 plays an important role in your results, should you give it more space in Section 2.3?

**Authors:** we improved the description of $SSB_0$ and parameter k related to $SSB_0$ on p.9 l.201-204 by changing "$SSB_0$ is the position of the inflection point, where the rate of recruitment production is highest, and k is the function's steepness at $SSB_0$" to the following "Parameter $SSB_0$ is the position of the inflection point of the recruitment abundance function. Parameter k is the steepness of the curve at point $SSB_0$, which defines the type of recruitment production function, which can be either purely compensatory ($k < 2$) or with a depensatory region ($k > 2$). In the latter case corresponding recruitment per capita function ($H(SSB_N)/SSB_N$) would have a minimum, that indicates the presence of Allee effect with $SSB_0$ in this case being an Allee effect threshold".

**R2-28**: 673 Use Equ./Eq. consistently.

**Authors:** all made consistent with "Eq".

**R2-29**: 680 What does 'q' stand for?

**Authors:** q stands for log-likelihood of the  model for which AIC is calculated and is given by the Eq. D1.

**R2-30**: Figure A1 This is a great illustration. However, it is somewhat challenging to distinguish the alphabets and arrows from the background. Could you use a color for the symbols?

**Authors:** Thank you for the suggestion. The colors have been changed.

**R2-31**: 565 'The' --> 'the'

**Authors:** corrected.

**R2-32**: Figure C1: This is a nice illustration. However, please tell in all subplots, to which parameter values the different curves correspond, similar to the subplot on the right lower corner.

**Authors:** The scheme is improved with parameter values corresponding to different curves.

[Figure]

**R2-33**: Figure D2: What are N, NFT etc. abbreviations of?

**Authors:** These were the intermediate model abbreviations, which were changed later to spnum, spwe and SST. Thank you for noticing, we have corrected Figure D2 with corresponding abbreviations.